# Unprocessing Seven Years of Algorithmic Fairness

**André F. Cruz**\* **& Moritz Hardt**
Max Planck Institute for Intelligent Systems, Tübingen and Tübingen AI Center

## Abstract

Seven years ago, researchers proposed a postprocessing method to equalize the error rates of a model across different demographic groups. The work launched hundreds of papers purporting to improve over the postprocessing baseline. We empirically evaluate these claims through thousands of model evaluations on several tabular datasets. We find that the fairness-accuracy Pareto frontier achieved by postprocessing the predictor with highest accuracy contains all other methods we were feasibly able to evaluate. In doing so, we address two common methodological errors that have confounded previous observations. One relates to the comparison of methods with different unconstrained base models. The other concerns methods achieving different levels of constraint relaxation. At the heart of our study is a simple idea we call unprocessing that roughly corresponds to the inverse of postprocessing. Unprocessing allows for a direct comparison of methods using different underlying models and levels of relaxation.

## 1 Introduction

Risk minimizing predictors generally have different error rates in different groups of a population. When errors are costly, some groups therefore seem to bear the brunt of uncertainty, while others enjoy the benefits of optimal prediction. This fact has been the basis of intense debate in the field of algorithmic fairness, dating back to the 1950s (Hutchinson & Mitchell, 2019). A difference in error rates between groups, equally deserving of a resource, strikes many as a moral wrong (Angwin et al., 2016; Barocas et al., 2019).

Researchers have therefore proposed numerous algorithmic interventions to mitigate a disparity in error rates. The most basic such method is known as postprocessing. Postprocessing sets group-specific acceptance thresholds so as to minimize risk while achieving an equality in error rates across a desired set of groups. Postprocessing is both simple and computationally efficient.

Perhaps because of its simplicity, postprocessing has been widely assumed to be sub-optimal. Troves of academic contributions seek to improve over postprocessing by more sophisticated algorithmic means. These efforts generally fall into two categories. Preprocessing methods aim to adjust the source data in such a manner that predictors trained on the data satisfy certain properties. So-called "inprocessing" methods, in contrast, modify the training algorithms itself to achieve a desired constraint during the optimization step.

### 1.1 Our contributions

Through a large, computationally intensive meta study we empirically establish that postprocessing is Pareto-dominant among all methods we were feasibly able to evaluate. Whatever level of accuracy can be achieved by any method at a specific level of error rate disparity, can also be achieved by setting group-specific acceptance thresholds on an unconstrained risk score.

We performed more than ten thousand model training and evaluation runs across five different prediction tasks from the `folktables` package (Ding et al., 2021), with two or four sensitive groups, based on tabular data from the US Census American Community Survey. We also include additional experiments on the Medical Expenditure Panel Survey (MEPS) (Blewett et al., 2021) dataset

---

\*Corresponding author: `andre.cruz@tuebingen.mpg.de`

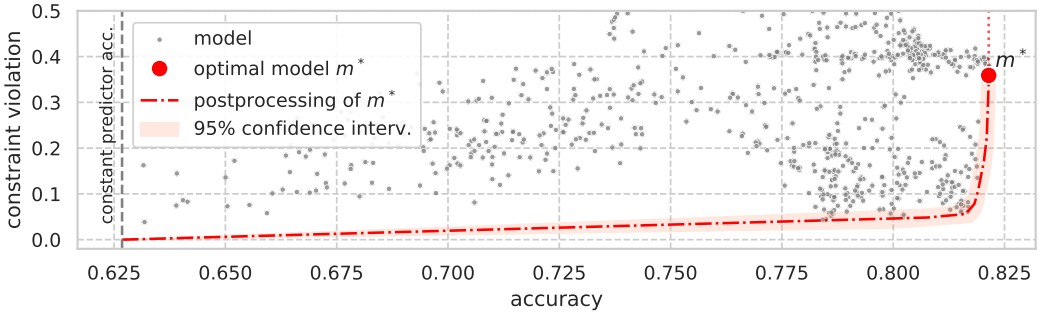

Figure 1: Test accuracy and constraint violation for 1000 models trained on the ACSIncome dataset (Ding et al., 2021), corresponding to a variety of preprocessing or inprocessing methods, as well as unconstrained learners. A red line shows the postprocessing Pareto frontier of the *single* model with highest accuracy (a GBM model).

in the appendix. The methods include recent state-of-the-art algorithms, as well as standard baselines. While postprocessing is hyperparameter-free, we did extensive search for the best hyperparameters of all competing methods.

Our work addresses two common methodological errors that have confounded previous comparisons with postprocessing.

First, many preprocessing and inprocessing methods naturally do not achieve exact error rate equality, but rather some relaxation of the constraint. In contrast, postprocessing is typically applied so as to achieve exact equality. The primary reason for this seems to be that there is a simple and efficient method based on tri-search to achieve exact equality (Hardt et al., 2016). However, an efficient relaxation of error rate parity is more subtle and is therefore lacking from popular software packages. We contribute a linear programming formulation to achieve approximate error rate parity for postprocessing, and open-source our implementation in an easy-to-use Python package called `error-parity`.[1] This allows us to compare methods to postprocessing at the same level of slack.

Second, different methods use base models of varying performance. Observed improvements may therefore be due to a better unconstrained base model rather than a better way of achieving error rate parity. How can we put different methods on a level playing field? We introduce a simple idea we call *unprocessing* that roughly corresponds to the inverse of postprocessing. Here, we take a model that satisfies error rate parity (approximately) and optimize group-specific thresholds so as to yield the best *unconstrained* model possible. Unprocessing maps any fairness-constrained model to a corresponding unconstrained counterpart. Both models have the same underlying risk-score estimates, to which we can then apply postprocessing. When comparing postprocessing to any given method we therefore do not have to come up with our own unconstrained model. We can simply steal, so to say, the unconstrained model implicit in any method.

These findings should not come as a surprise. Theory, perhaps overlooked, had long contributed an important fact: If an unconstrained predictor is close to Bayes optimal (in squared loss), then postprocessing this predictor is close to optimal among all predictors satisfying error rate parity (Hardt et al., 2016, Theorem 4.5). To be sure, this theorem applies to the squared loss and there are clever counterexamples in some other cases (Woodworth et al., 2017). However, our empirical evaluation suggests that these counterexamples don't arise in the real datasets we considered. This may be the case because, on the tabular datasets we consider, methods such as gradient boosting produce scores that are likely close to Bayes optimal under the squared loss. We focus on tabular data case-studies, an important basis for public policy decisions. High-dimensional datasets containing raw features (e.g., images, text) are not explored in the current paper, making for an interesting future work direction.

**Limitations and broader impacts.** We are narrowly concerned with evaluating algorithms that achieve error rate parity approximately. We do not contribute any new substantive insights about fairness in different domains. Nor do we escape the many valid criticisms that have been brought

---

[1] https://github.com/socialfoundations/error-parity

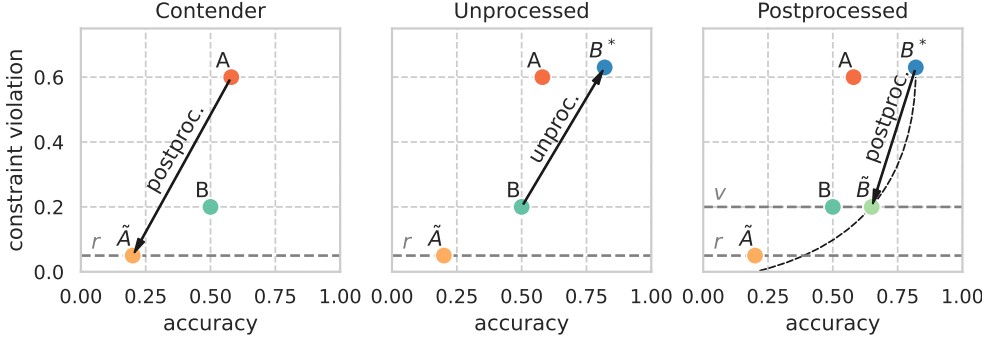

Figure 2: Example illustrating unprocessing. Left: Initial unconstrained model $A$ postprocessed to $\tilde{A}$. Some contender model $B$ incomparable to $\tilde{A}$. Middle: We unprocess $B$ to get a new model $B^*$. Right: Postprocessing $B^*$ to the same constraint level as $B$ or $\tilde{A}$.

forward against algorithmic fairness narrowly construed (Bao et al., 2021; Barocas et al., 2019; Kasy & Abebe, 2021). In particular, our work says nothing new about the question whether we should equalize error rates in the first place. Some argue that error rates should be a diagnostic, not a locus of intervention (Barocas et al., 2019). Others reject the idea altogether (Corbett-Davies et al., 2017). If, however, the goal is to equalize error rates exactly or approximately, the simplest way of doing so is optimal: *Take the best available unconstrained model and optimize over group-specific thresholds.*

### 1.2 RELATED WORK

Hardt et al. (2016) introduced error rate parity under the name of *equalized odds* in the context of machine learning and gave an analysis of postprocessing, including the aforementioned tri-search algorithm and theoretical fact. Woodworth et al. (2017) proposed a second-moment relaxation of equalized odds and an algorithm to achieve the relaxation on linear predictors, as well as examples of specific loss functions and predictors for which postprocessing is not optimal. The postprocessing fairness-accuracy trade-off has since been further detailed (Jang et al., 2022; Kim et al., 2020).

Numerous works have considered constrained empirical risk formulations to achieve fairness criteria, see, e.g., Agarwal et al. (2018); Celis et al. (2019); Cotter et al. (2019); Cruz et al. (2023); Donini et al. (2018); Menon & Williamson (2018); Zafar et al. (2017; 2019) for a starting point. The work on learning fair representations (Zemel et al., 2013) spawned much follow-up work on various preprocessing methods. See Section 2 for an extended discussion of the related work that we draw on in our experiments. We are unable to survey the vast space of algorithmic fairness methods here.

Our findings mirror several studies in ML and related fields that advocate for increased empirical rigor when proposing complex developments over simple baselines. Armstrong et al. (2009), and later Kharazmi et al. (2016), present evidence of the widespread use of weak baselines in information retrieval, leading to over-stated advancements in the field. Lucic et al. (2018) study state-of-the-art generative adversarial networks (GAN), and find no clear-cut improvements over the original GAN introduced in Goodfellow et al. (2014). Ferrari Dacrema et al. (2019) reach similar conclusions for the field of recommenders systems, and Musgrave et al. (2020) for the field of metric learning. We contribute to this growing body of work by showing that a simple postprocessing baseline matches or dominates all evaluated fairness interventions over a variety of datasets and evaluation scenarios.

## 2 EXPERIMENTAL SETUP

We conduct experiments on four standard machine learning models, paired with five popular algorithmic fairness methods. The standard unconstrained models in the comparison are: gradient boosting machine (GBM), random forest (RF), neural network (NN), and logistic regression (LR).

Regarding fairness interventions, we include both pre- and inprocessing methods in our experiments. We use the learned fair representations (LFR) (Zemel et al., 2013) and the correlation remover

(CR) (Bird et al., 2020) preprocessing fairness methods, respectively implemented in the `aif360` and `fairlearn` Python libraries. Additionally, we use the exponentiated gradient reduction (EG) and the grid search reduction (GS) inprocessing fairness methods (Agarwal et al., 2018) (implemented in `fairlearn`), as well as the FairGBM (Cruz et al., 2023) inprocessing method (implemented in `fairgbm`). The preprocessing methods (CR, LFR) can be paired with any other ML model ($2 \cdot 4 = 8$ pairs), EG as well (4 pairs), and GS is compatible with GBM, RF, and LR models (3 pairs). FairGBM is naturally only compatible with GBM. Together with the four standard unconstrained models, there is a total of 20 different methods (or pairings) in the comparison.

Although there have been numerous proposed fairness methods over the years, far fewer have available and ready-to-use open-source implementations. This is somewhat inevitable, as each new method would have to maintain a usable up-to-date implementation, as well as custom implementations for compatibility with different fairness criteria and different underlying base learners. Postprocessing approaches have a practical advantage: a single implementation is compatible with any underlying learner that can produce scores of predicted probabilities, and any fairness criterion that can be expressed as a constraint over the joint distribution of $(Y, \hat{Y}, S)$, where $Y$ is the true target, $\hat{Y}$ the predictions, and $S$ the protected group membership.

On each dataset, we train 50 instances of each ML algorithm in the study. For clarity, we will refer to different pairs of ⟨unconstrained, fairness-aware⟩ algorithms as different algorithms (e.g., ⟨GBM, EG⟩ and ⟨NN, EG⟩ are two different algorithms). As we study 20 different ML algorithms, a total of $50 \cdot 20 = 1000$ ML models is trained on each dataset. Each model is trained with a different randomly-sampled selection of hyperparameters (e.g., learning rate of a GBM, number of trees of an RF, weight regularization of an LR). This fulfills two goals: first, to accurately explore the best outcomes of competing fair ML methods, as related work has shown that a wide range of fairness values can be obtained for the same ML algorithm by simply varying its hyperparameters; and, second, to indirectly benchmark against fairness-aware AutoML approaches, which attempt to train fair models by tuning the hyperparameters of unconstrained models (Cruz et al., 2021; Perrone et al., 2021; Weerts et al., 2023).

**Unprocessing.** We define $\pi_r(f)$ as the process of postprocessing a predictor $f$ to minimize some classification loss function $\ell$ over the group-specific decision thresholds $t_s \in \mathbb{R}$, subject to an $r$-relaxed equalized odds constraint (Equation 1),

$$\max_{y \in \{0,1\}} \left( \mathbb{P}[\hat{Y} = 1 | S = a, Y = y] - \mathbb{P}[\hat{Y} = 1 | S = b, Y = y] \right) \leq r, \quad \forall a, b \in \mathcal{S}, \qquad (1)$$

where the prediction for a sample of group $s \in \mathcal{S}$ is given by $\hat{Y} = \mathbb{1}\{\hat{R} \geq t_s\}$, and $\hat{R}$ is its real-valued risk score. Thereby, *unprocessing* is defined as the unconstrained minimization of the loss $\ell$; i.e., $\pi_\infty(f)$, an $\infty$-relaxed solution to equalized odds. As classifiers with different values of constraint violation are potentially incomparable between themselves, *unprocessing* emerges as a means to fairer comparisons between classifiers, unearthing the achievable unconstrained accuracy underlying a constrained predictor. For example, while classifiers $A$, $B$, and $\tilde{A}$ of Figure 2 are all Pareto-efficient (Pareto, 1919) (i.e., incomparable), we can fairly compare the accuracy of $A$ with that of $\pi_\infty(B) = B^*$ (both are unconstrained classifiers), $B$ with $\pi_v(B) = \tilde{B}$, and $\tilde{A}$ with $\pi_r(B)$.

The following subsections will detail the datasets we use (Section 2.1), and the experimental procedure we employ to test our hypothesis (Section 2.2).

## 2.1 DATASETS

We evaluate all methods on five large public benchmark datasets from the `folktables` Python package (Ding et al., 2021). These datasets are derived from the American Community Survey (ACS) public use microdata sample from 2018, containing a variety of demographic features (e.g., age, race, education). We also conduct a similar experiment on the MEPS dataset, shown in Appendix A.5.

Each of the five ACS datasets is named after a specific prediction task: ACSIncome (1.6M rows) relates to household income prediction, ACSTravelTime (1.4M rows) relates to daily commute time prediction, ACSPublicCoverage (1.1M rows) relates to health insurance coverage prediction, ACSMobility (0.6M rows) relates to the prediction of address changes, and ACSEmployment (2.3M rows) relates to employment status prediction. ACSIncome arguably carries particular weight in

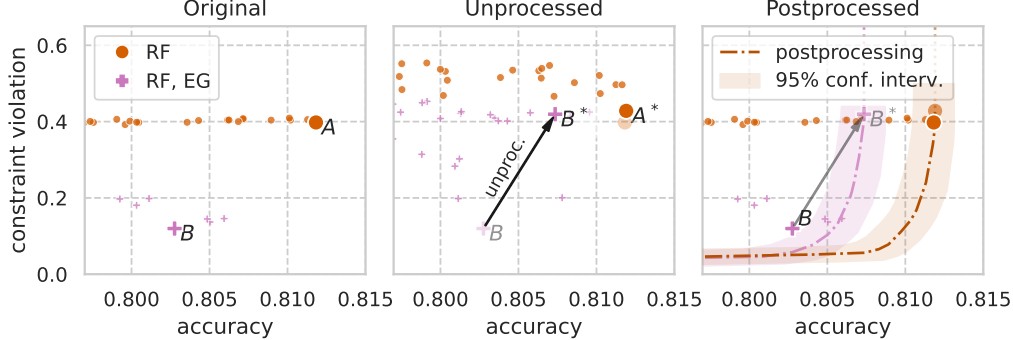

Figure 3: Real-data version of the illustrative plot shown in Figure 2 (results shown on ACSIncome test, models selected on validation). $A$ and $B$ are two arbitrary incomparable models (both Pareto-efficient), which are made comparable after unprocessing. *Left*: original (unaltered) results; *Middle*: results after unprocessing all models; *Right*: original (unaltered) results, together with the postprocessing curve for both $A^*$ and $B^*$. Additional model pairs shown in Appendix A.6.

the fair ML community, as it is a larger modern-day version of the popular UCI Adult dataset (49K rows) (Dua & Graff, 2017), which has been widely used for benchmarking algorithmic fairness methods over the years. We use race group membership as the protected attribute on all five datasets (RAC1P column); specifically, we use samples from the four largest groups: *White*, *Black*, *Asian*, and *Other* (some other race alone). Additional experiments using only samples from the two largest groups are presented as well, although not the focus of the paper results' analysis (see Appendix A.4).

In total, 11 000 models were trained and evaluated over a range of 11 different evaluation scenarios, pertaining to 6 datasets, with sizes ranging from 49K to 2.3M samples.

## 2.2 Experimental procedure

We conduct the following procedure for each dataset, with a $60\%/20\%/20\%$ train/test/validation data split. First, we fit 1000 different ML models on the training data (50 per algorithm type). Second, to enable comparison of all models on an equal footing, we unprocess all 1000 trained models (on validation), and compute accuracy and equalized odds violation of the resulting classifiers. For any given classifier, its equalized odds violation is given by the left-hand side of the inequality in Equation 1 (or the smallest slack $r$ that fulfills the inequality). Then, we select the model with highest *unprocessed* accuracy, $m^* = \pi_\infty(m')$, obtained by the unconstrained postprocessing of the model $m'$. We defer the formal definition and procedure for solving the relaxed problem to Section 4.

We solve the $r$-relaxed equalized odds postprocessing on validation, $\pi_r(m')$, for all values of constraint violation, $r \in [0, c(m^*)]$ (with discrete intervals of 0.01), where $c(m^*)$ is the constraint violation of $m^*$. Finally, we compute accuracy and equalized odds violation on the withheld test dataset for all original 1000 models, and all post-processed versions of $m'$, $\pi_r(m')$. All in all, even though the selection process for $m^*$ and $m'$ entirely disregarded fairness, we expect $\pi_r(m')$ to be the classifier with highest accuracy at all levels of fairness $r \in [0.0, 1.0]$ — as illustrated in Figure 2.

It may happen that the classifier $m^*$ with highest unprocessed accuracy is based on a pre- or inprocessing fairness method $m'$. Crucially, this implies that the means by which this fairness method resulted in a fairer classifier was by finding more accurate risk scores in the first place. Otherwise, unprocessing fairness-constrained models would not result in accurate unconstrained predictions.

Figure 3 shows an example of the general effect of unprocessing on the ACSIncome dataset (test results). Example models $A$ and $B$ are chosen respectively to maximize accuracy ($A$) and to maximize an average of accuracy and fairness ($B$) on validation data. On the left plot (original, unaltered models), unconstrained (● markers) and constrained models (✚ markers) are incomparable in terms of Pareto dominance: the first is generally more accurate than the latter, but the ranking is reversed for constraint violation. However, after unprocessing (middle plot), we can see that the unprocessed unconstrained model $A$ achieves higher accuracy than the unprocessed constrained

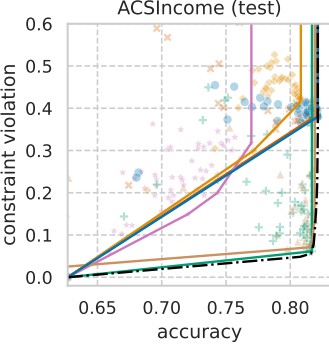 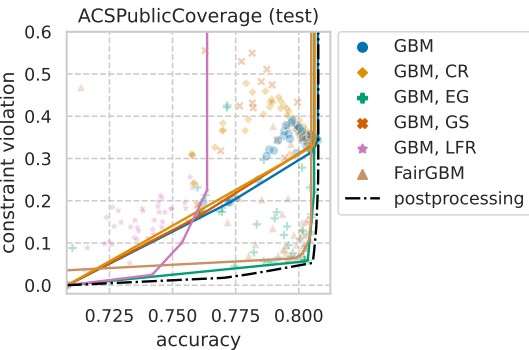

Figure 4: Pareto frontier attained by each GBM-based algorithm, together with the Pareto frontier attained by postprocessing the GBM-based model with highest unprocessed validation accuracy, $m^*$. Results for remaining ACS datasets shown in Figure A1.

model $B$, indicating that postprocessing the first would be Pareto dominant or match the latter. Finally, the right plot confirms this hypothesis as evident by comparing postprocessing curves for both models. More extensive experimental results are shown in Section 3.

## 3 RESULTS ON AMERICAN COMMUNITY SURVEY DATA

In this section we will present and discuss the results of experiments on all five ACS datasets. These experiments entail a total of 1 000 models trained per dataset. Due to space constraints, plots are shown only for the ACSIncome and ACSPublicCoverage datasets. Corresponding plots for the remaining datasets are shown in Appendix A. Results for a counterpart experiment using only two sensitive groups are also explored in this section, and further detailed in Appendix A.4.

### 3.1 COMPARISON BETWEEN FAIRNESS METHODS

We first analyze how each pre- or inprocessing fairness method compares with each other, without the effects of postprocessing. Figure 4 shows the Pareto frontiers achieved by each method when using GBM base models (see also Figure A1 of the appendix). Overall, preprocessing methods (LFR and CR) achieved lacklustre fairness-accuracy trade-offs across all datasets, while the EG and FairGBM inprocessing methods performed best (highest area above Pareto frontiers). Specifically, LFR is indeed able to achieve high fairness fulfillment, but at a steep accuracy cost. To clarify: the plotted colored Pareto frontiers correspond to *multiple* (up to 50) different underlying base models, while the black dashed line corresponds to the postprocessings of the *single* GBM-based model with highest accuracy. The following subsection contains a more detailed analysis of postprocessing.

Interestingly, some fairness methods were able to achieve higher *test* accuracy than unconstrained GBM models, suggesting improved generalization performance. Figure 5 shows one potential reason: fairness methods can take notoriously high compute resources to train, potentially giving them a compute advantage with respect to their unconstrained counterparts. Recent related work has put forth other explanations for why fairness-constraining can improve learning. Wei et al. (2023) find that fairness can improve overall performance under label noise by improving learning on tail sub-populations. On the other hand, Creager et al. (2021) establish ties between common fairness constraints and goals from the robustness literature, suggesting that fairness-aware learning can improve generalization under distribution shifts.

### 3.2 POSTPROCESSING VS OTHER METHODS

Figure 6 shows test-set results for the experimental procedure detailed in Section 2.2, zoomed on the region of interest (high accuracy and low constraint violation). The model with highest *unprocessed* validation accuracy, $m^*$, is shown with a larger marker, while all other markers correspond to

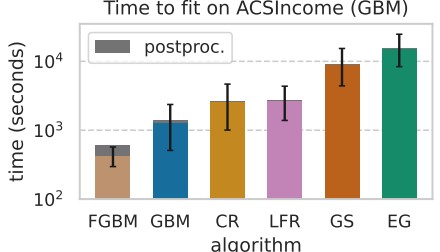 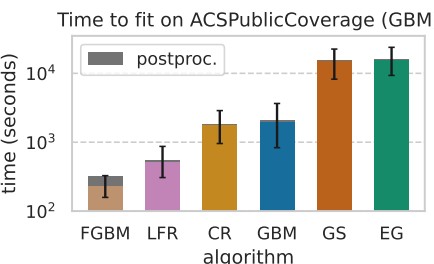

Figure 5: Mean time to fit GBM and GBM-based preprocessing and inprocessing algorithms on the ACSIncome (left plot) and ACSPublicCoverage (right plot) datasets, with 95% confidence intervals. The time taken to run postprocessing is also shown for each algorithm as a stacked dark bar. Note the *log* scale: the EG inprocessing method takes one order of magnitude longer to fit than the base GBM.

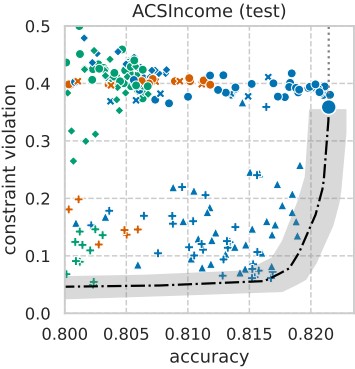 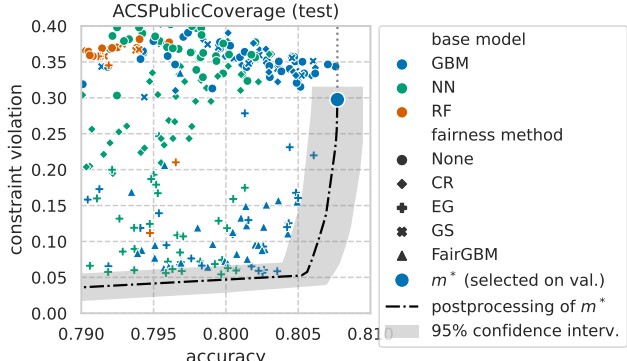

Figure 6: Detailed look at the postprocessing Pareto frontier on the ACSIncome (left) and ACSPublic-Coverage (right) datasets. As shown, postprocessing $m^*$ dominates or matches all 1000 trained ML models, regardless of the underlying train algorithm (preprocessing, inprocessing, or unconstrained). Results for remaining ACS datasets shown in Figure A7.

unaltered non-postprocessed models. The Pareto frontier achieved by postprocessing $m^*$ is shown with a black dash-dot line, as well as corresponding 95% confidence intervals computed using bootstrapping (Efron & Tibshirani, 1994). Figures A2–A6 show a wider view of the same underlying data, as some algorithms fail to show-up in the zoomed region of interest. Figures A8–A12 show the result of conducting the same experiment but using only GBM-based models, leading to identical trends.

All datasets show a wide spread of models throughout the fairness-accuracy space, although to varying levels of maximum accuracy (from 0.713 on ACSTravelTime to 0.831 on ACSEmployment). Unconstrained models (circles) can generally be seen to form a cluster of high accuracy and low fairness (high constraint violation). Neither LFR nor LR-based methods manage to produce any model within the region plotted in Figure 6. On the ACSIncome and ACSPublicCoverage datasets, the $m^*$ model corresponds to an unconstrained GBM (blue circle), on ACSTravelTime and ACSEmployment $m^*$ is of type ⟨GBM, CR⟩ (blue diamond), and on ACSMobility $m^*$ is of type ⟨GBM, GS⟩ (blue cross). Models $m^*$ are GBM-based across all datasets, contributing to a wide body of literature reporting that GBM models are highly performant on tabular datasets (Shwartz-Ziv & Armon, 2022).

Crucially, postprocessing the *single* most accurate model resulted in the fair optima for all values of fairness constraint violation on all datasets, either dominating or matching other contender models (within 95% confidence intervals). That is, all optimal trade-offs between fairness and accuracy can be retrieved by applying different group-specific thresholds to the same underlying risk scores.

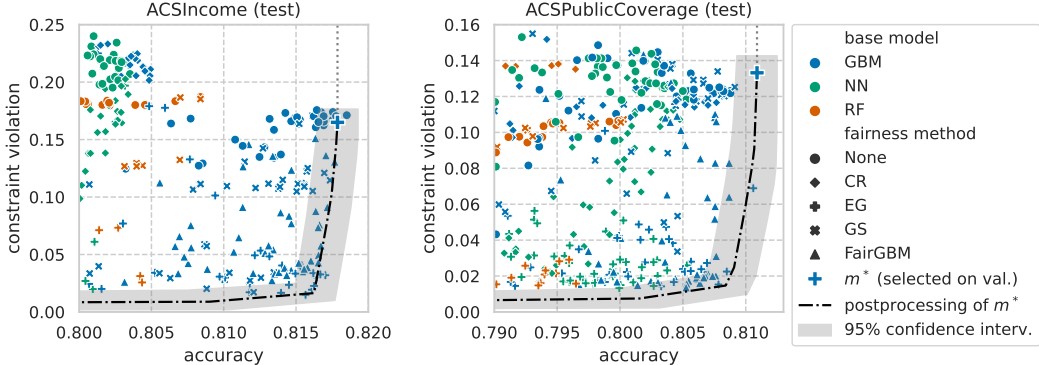

Figure 7: [**Binary protected groups**] Results for a counterpart to the main experiment, in which only samples from the two largest groups are used (*White* and *Black*). Note the significantly reduced $y$ axis range when compared with Figure 6. Results for remaining ACS datasets shown in Figure A14.

Finally, Figure 7 shows results for a similar experiment where fairness constraints were learned only on the two largest sub-groups (*White* and *Black*). This leads to an arguably easier problem to solve, which is reflected on the general compression of models on the vertical axis (reduced constraint violation for all models). While previously the maximum unprocessed accuracy on ACSIncome was achieved at $0.38$ constraint violation, on this binary-group setting it is achieved at $0.16$ constraint violation. Nonetheless, the same trend is visible on all studied datasets. Unconstrained models — either trained in an unconstrained manner (circles) or made unconstrained via unprocessing ($m^*$) — occupy regions of high accuracy and low fairness (high constraint violation). However, the Pareto frontier that results from postprocessing $m^*$ (the best-performing unconstrained model) to different levels of fairness relaxation again dominates or matches the remaining fairness methods.

All in all, postprocessing provides a full view of the Pareto frontier derived from a single predictor $m^*$. Regardless of fairness violation, when this predictor is near-optimal — potentially achievable on tabular data by training a variety of algorithms — so will its postprocessed Pareto frontier be.

## 4 ACHIEVING RELAXED ERROR RATE PARITY

Error rate parity, also known as equalized odds, enforces equal false positive rate (FPR) and equal true positive rate (TPR) between different protected groups (Hardt et al., 2016). This can be formalized as a constraint on the joint distribution of $(Y, \hat{Y}, S)$:

$$\mathbb{P}[\hat{Y} = 1 | S = a, Y = y] = \mathbb{P}[\hat{Y} = 1 | S = b, Y = y], \quad \forall y \in \{0, 1\}, \quad \forall a, b \in \mathcal{S}, \qquad (2)$$

where $a \neq b$ references two distinct groups in the set of all possible groups $\mathcal{S}$.

Fulfilling the strict equalized odds constraint greatly simplifies the optimization problem of finding the optimal classifier through postprocessing, as the constrained optimum must be at the intersection of the convex hulls of each group-specific ROC curve. As such, we're left with a linear optimization problem on a single 2-dimensional variable, $\gamma = (\gamma_0, \gamma_1)$:

$$\min_{\gamma \in D} \gamma_0 \cdot \ell(1, 0) \cdot p_0 + (1 - \gamma_1) \cdot \ell(0, 1) \cdot p_1, \qquad (3)$$

where $\gamma_0$ is the global FPR, $\gamma_1$ the global TPR, $D \subset [0, 1]^2$ the optimization domain, $p_y = \mathbb{P}[Y = y]$ the prevalence of label $Y = y$, and $\ell(\hat{y}, y)$ the loss incurred for predicting $\hat{y}$ when the correct class was $y$ (we assume w.l.o.g. $\ell(0, 0) = \ell(1, 1) = 0$). Strict equalized odds fulfillment collapses the optimization domain $D$ into a single convex polygon that results from intersecting all group-specific ROC hulls; i.e., $D = \bigcap_{s \in \mathcal{S}} D_s$, where $D_s$ is the convex hull of the ROC curve for group $s$. Specifically, $D_s = \text{convexhull}\{C_s(t) : t \in \mathbb{R}\}$, and $C_s$ defines the ROC curve for group $s$ as:

$$C_s(t) = \left( \mathbb{P}\left[ \hat{R} \geq t | S = s, Y = 0 \right], \mathbb{P}\left[ \hat{R} \geq t | S = s, Y = 1 \right] \right), \qquad (4)$$

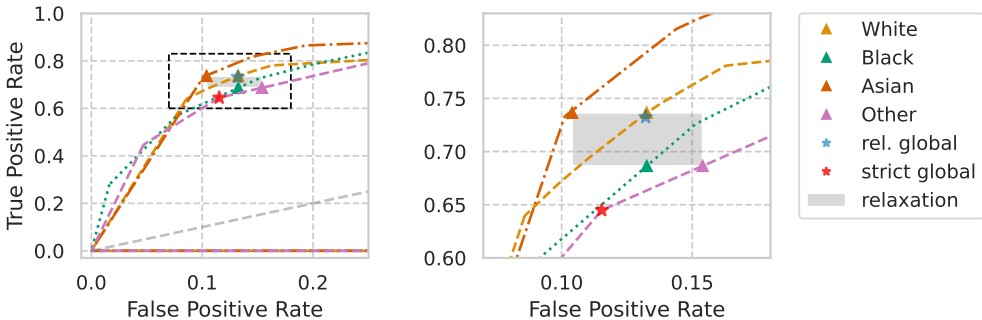

Figure 8: Optimal solution to a strict (red star) and a 0.05-relaxed (blue star) equalized odds constraint. Right plot shows a zoom on the region of interest (represented on the left by dashed black rectangle).

where $t \in \mathbb{R}$ is a group-specific decision threshold, and $\hat{R}$ is the predictor's real-valued score.

In this section, we detail the solution to the $r$-relaxed equalized odds constraint defined in Equation 1. In order to relax the equalized odds constraint, we introduce a slack variable, $\delta^{(a,b)} \in [0,1]^2$, for each pair $a, b \in \mathcal{S}$:

$$\mathbb{P}[\hat{Y} = 1 | S = a, Y = y] - \mathbb{P}[\hat{Y} = 1 | S = b, Y = y] = \delta_y^{(a,b)} \leq r, \quad \forall y \in \{0,1\}, \quad (5)$$

where $r \in [0,1]$ is the maximum allowed constraint violation.

We introduce variables $\gamma^{(s)} = (\gamma_0^{(s)}, \gamma_1^{(s)}) \in D_s, s \in \mathcal{S}$, as the points of group-specific FPR, $\gamma_0^{(s)}$, and group-specific TPR, $\gamma_1^{(s)}$. Equation 5 can then be equivalently stated as:

$$\left\| \gamma^{(a)} - \gamma^{(b)} \right\|_\infty = \left\| \delta^{(a,b)} \right\|_\infty \leq r. \quad (6)$$

The global ROC point, $\gamma$, is defined as:

$$\gamma_0 = \sum_{s \in \mathcal{S}} \gamma_0^{(s)} \cdot p_{s|0}, \qquad \gamma_1 = \sum_{s \in \mathcal{S}} \gamma_1^{(s)} \cdot p_{s|1}, \quad (7)$$

where $p_{s|y} = \mathbb{P}[S = s | Y = y]$ is the relative size of group $s$ within the set of samples with label $Y = y$. Importantly, the global point $\gamma$ is not limited to the intersection of group-specific ROC hulls. Each group-specific ROC point is naturally limited to be inside its group-specific ROC hull, $\gamma^{(s)} \in D_s$, and $\gamma$ is only limited by its definition as a function of all $\gamma^{(s)}, s \in \mathcal{S}$, as per Equation 7.

Finally, finding the $r$-relaxed optimum boils down to minimizing the linear objective function defined in Equation 3, with domain $D = \bigcup_{s \in \mathcal{S}} D_s$, subject to affine constraints defined in Equations 6–7. This optimization problem amounts to a linear program (LP), for which there is a variety of efficient open-source solvers (Diamond & Boyd, 2016). We contribute a solution in an open-source package.[1]

Figure 8 shows an example of optimal strict and 0.05-relaxed solutions for equalized odds. Strict fulfillment of the equalized odds constraint (red star) reduces the feasible space of solutions to the intersection of all group-specific ROC hulls. This fact potentially restricts all but one group to sub-optimal accuracy, achieved by randomizing some portion of the classifier's predictions. On the other hand, if we allow for some relaxation of the constraint, each group's ROC point will lie closer to its optimum. In this example, the optimal solution to an $r = 0.05$ relaxation no longer needs to resort to randomization, placing each group's ROC point on the frontier of its ROC convex hull.

## 5 CONCLUSION

We revisit the simple postprocessing method in a comprehensive empirical evaluation spanning 6 distinct datasets, 11 evaluation tasks, and more than $11\,000$ trained models. We find that, in all cases, any Pareto-optimal trade-off between accuracy and error rate parity can be achieved by postprocessing the model with highest accuracy. Along the way, we address two confounding factors that have impaired previous comparisons of fairness methods. We hope that our study helps strengthen evaluation standards in algorithmic fairness.

## ACKNOWLEDGMENTS

We're indebted to Noam Barda, Noa Dagan, and Guy Rothblum for insightful and stimulating discussions about the project. We thank Florian Dorner, Olawale Salaudeen, and Vivian Nastl for invaluable feedback on an earlier version of this paper. Lastly, we thank the four anonymous reviewers for their fruitful suggestions, and the area chair for important and enriching references to related work on the undue over-complexification of ML methods.

The authors thank the International Max Planck Research School for Intelligent Systems (IMPRS-IS) for supporting André F. Cruz.

## REPRODUCIBILITY STATEMENT

We've made significant efforts to ease reproducibility of our experiments. All source code has been open-sourced, including open-sourcing a Python package[1] to postprocess any score-based classifier to a given level of fairness-constraint relaxation, code to run experiments using the aforementioned package (folder `scripts` of the supplementary materials[2]), and code to generate the paper plots (folder `notebooks` of the supplementary materials[2]). Detailed hyperparameter search spaces for each algorithm are included in folder `hyperparameters_spaces` of the supplementary materials.[2] Furthermore, we are releasing detailed experimental results for all trained models in a series of `csv` files (under folder `results` of the supplementary materials[2]), including a variety of performance and fairness metrics, as well as their values at 2.5 and 97.5 bootstrapping percentiles. Appendix B details the infrastructure used to run all jobs, as well as total compute usage.

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

## A    ADDITIONAL EXPERIMENTAL RESULTS

The main body of the paper discusses results on all five ACS datasets. However, due to space constraints, plots are only shown for two example datasets: ACSIncome and ACSPublicCoverage. Appendices A.1–A.3 show analogous versions of each previous plot for the remaining three ACS datasets: ACSTravelTime, ACSMobility, and ACSEmployment. Plots are shown in the same order as in the main paper body. Additionally, we run a similar experiment on ACS datasets using only the two largest sensitive groups (*White* and *Black*), shown in Appendix A.4. Appendix A.5 presents results on the MEPS dataset (Blewett et al., 2021), an entirely different data source corresponding to real-world surveys of healthcare usage across the United States. Appendix A.6 provides further evidence that model ranking is maintained throughout all levels of constraint violation, and Appendix A.7 compares the results of unconstrained model training to unprocessing constrained models.

We consciously refrain from evaluating on the popular COMPAS dataset (Angwin et al., 2016), as related work has surfaced severe data gathering issues, including measurement biases and label leakage (Bao et al., 2021; Barenstein, 2019; Fabris et al., 2022). The German Credit dataset (Dua & Graff, 2017) — another popular benchmark in the fairness literature — suffers from its small size (1 000 samples), the age of its data (dates back to 1973–1975), and encoding issues that make it impossible to retrieve accurate sensitive information such as the individual's sex (Grömping, 2019). Overall, a total of 11 different evaluation scenarios were studied, pertaining to 6 datasets, with sizes ranging from 49K to 2.3M samples. Confidence intervals and metric results are computed using bootstrapping on the respective evaluation dataset (Efron & Tibshirani, 1994). We hope the scale of our study suffices to convince the reader of the validity of our claims. Source code is made available to easily reproduce our setup on other datasets.[2] All appendix experiments are in accordance with the main findings presented in Section 3.

### A.1    COMPARISON BETWEEN FAIRNESS METHODS

Figure A1 shows Pareto frontiers for all studied GBM-based algorithms. We observe a similar trend to that seen in Figure 4: preprocessing fairness methods can increase fairness but at dramatic accuracy costs, while EG and FairGBM inprocessing fairness methods trade Pareto-dominance between each other. Postprocessing Pareto frontier is also shown for reference, but a more detailed comparison between postprocessing and all other contender models is shown in the following section.

### A.2    POSTPROCESSING VS OTHER METHODS

Figures A2–A6 show complete views of the Pareto frontiers obtained by postprocessing the model with highest validation accuracy $m^*$ on each dataset (potentially obtained by unprocessing a fairness-aware model), together with a scatter of all other competing preprocessing, inprocessing, or unconstrained models (1 000 in total per dataset). Figure A7 shows detailed postprocessing results on each dataset, zoomed on the region of interest (maximal accuracy and minimal constraint violation, i.e., bottom right portion of the plot). Figures A8–A12 show results using only a subset of models: only GBM-based models. The main paper hypothesis is confirmed on each and every plot: we can obtain optimally fair classifiers at any level of constraint violation by postprocessing the model with highest accuracy, $m^*$, irrespective of its constraint violation.

---

[2]Supplementary materials:
https://github.com/socialfoundations/error-parity/tree/supp-materials

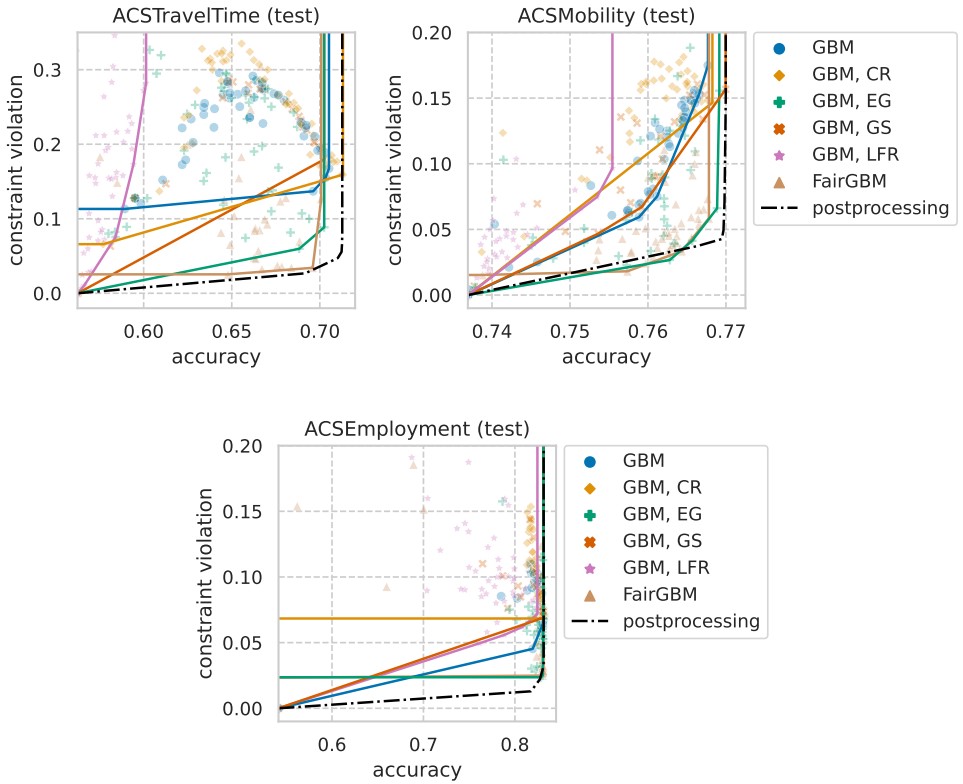

Figure A1: Pareto frontier attainable by each GBM-based ML algorithm, together with the Pareto frontier attained by postprocessing $m^*$, the GBM-based model with highest unprocessed validation accuracy. Plotted Pareto curves are linearly interpolated between Pareto-efficient models.

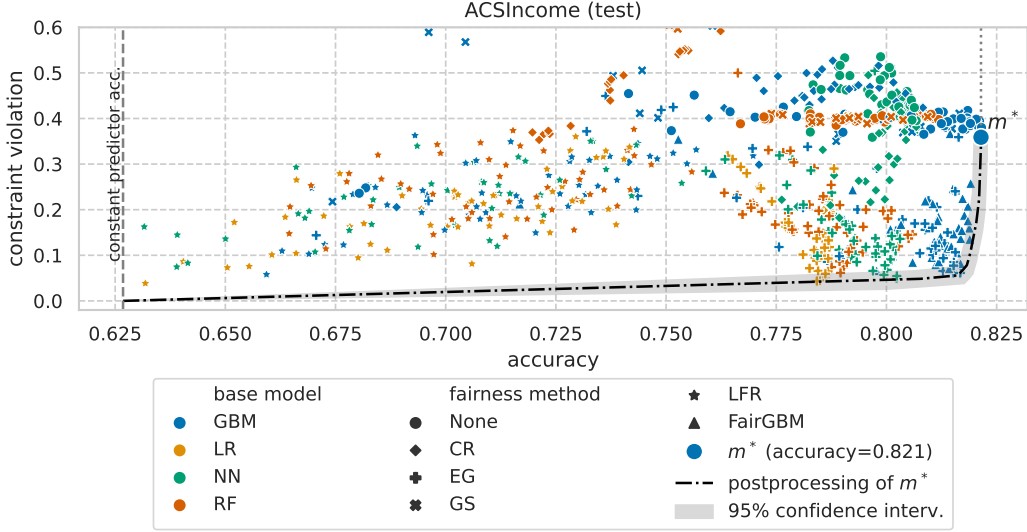

Figure A2: Fairness and accuracy test results for all $1\,000$ trained ML models (50 of each type) on the ACSIncome dataset. Colors portray different underlying unconstrained models and markers portray different fairness methods (or no fairness method for circle markers). The unconstrained model with highest validation accuracy, $m^*$, is shown with a larger marker, and the Pareto frontier attainable by postprocessing $m^*$ is shown as a black dash-dot line, together with its 95% confidence intervals in shade. This is a colored and more granular version of Figure 1.

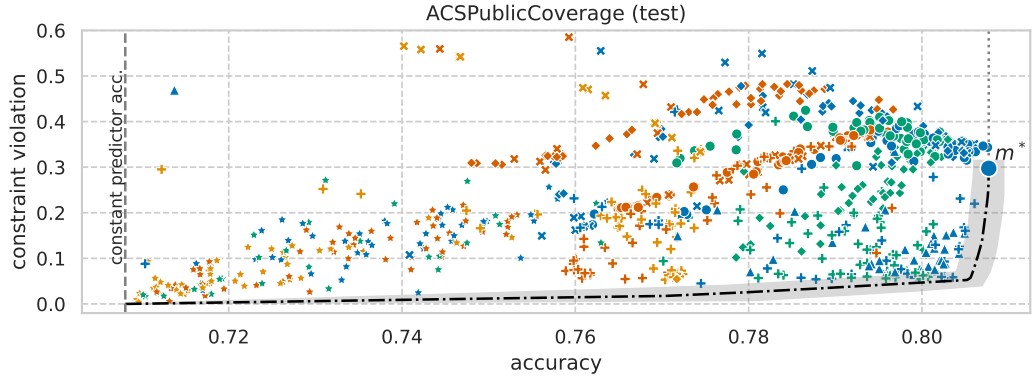

Figure A3: Fairness and accuracy test results on the ACSPublicCoverage dataset. Model $m^*$ is of type $\langle$GBM$\rangle$ and achieves $0.808$ accuracy. See legend and caption of Figure A2 for more details.

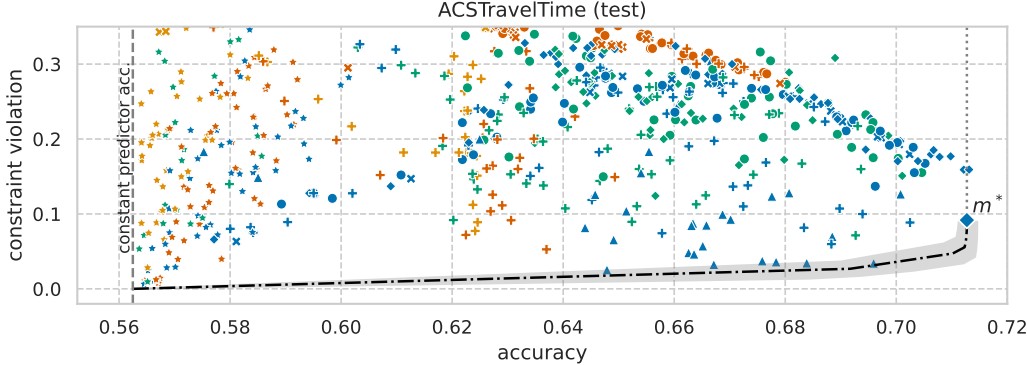

Figure A4: Fairness and accuracy test results on the ACSTravelTime dataset. Model $m^*$ is of type $\langle$GBM, CR$\rangle$ and achieves $0.713$ accuracy. See legend and caption of Figure A2 for more details.

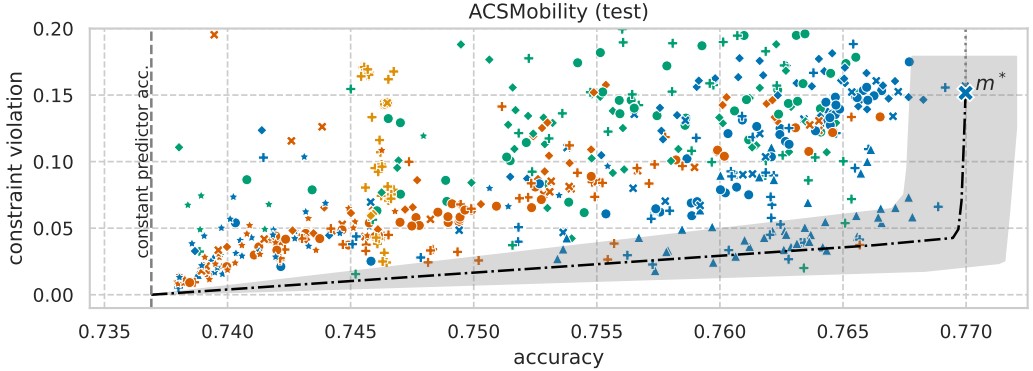

Figure A5: Fairness and accuracy test results on the ACSMobility dataset. Model $m^*$ is of type $\langle$GBM, GS$\rangle$ and achieves $0.770$ accuracy. See legend and caption of Figure A2 for more details.

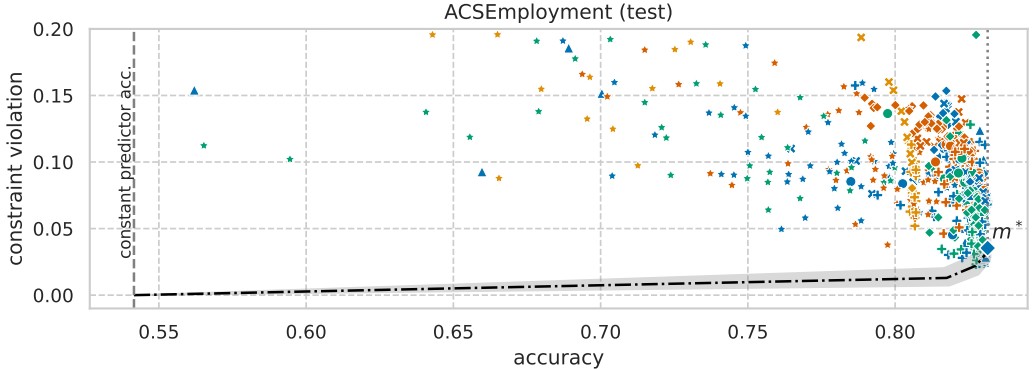

Figure A6: Fairness and accuracy test results on the ACSEmployment dataset. Model $m^*$ is of type $\langle$GBM, CR$\rangle$ and achieves $0.831$ accuracy. See legend and caption of Figure A2 for more details.

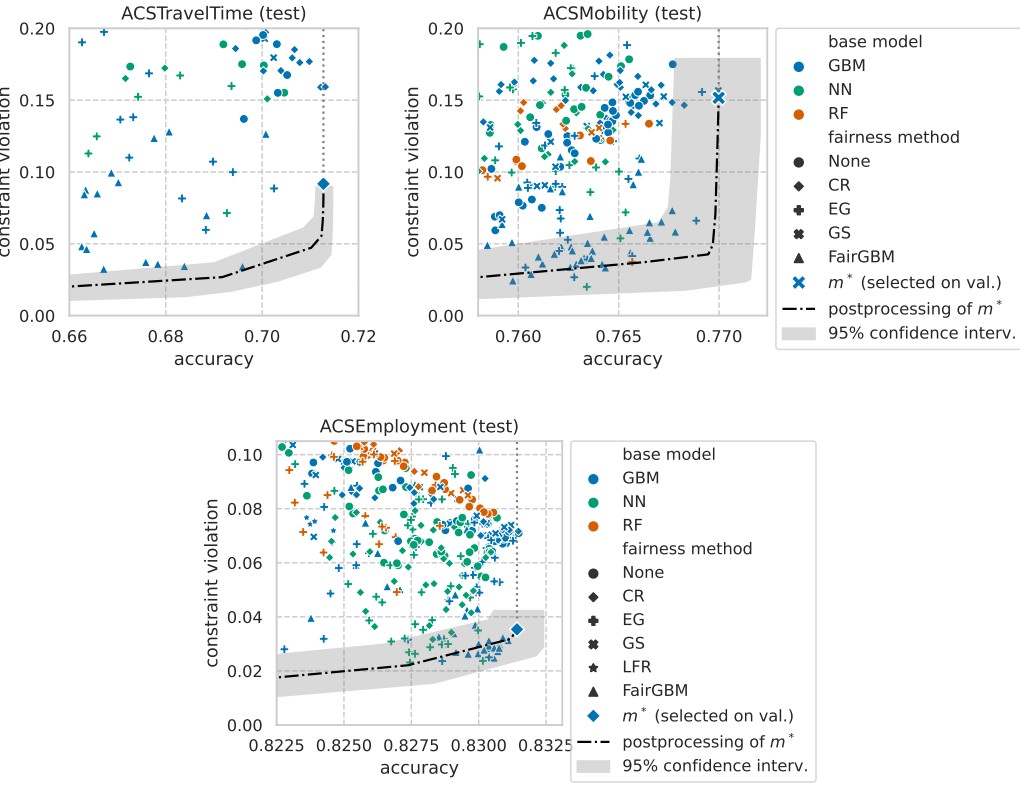

Figure A7: Detailed view of the postprocessing Pareto frontier on the ACSTravelTime (left), ACSMobility (right), and ACSEmployment (bottom) datasets. Respectively corresponds to zoomed-in versions of Figures A4 (left), A5 (right), and A6 (bottom).

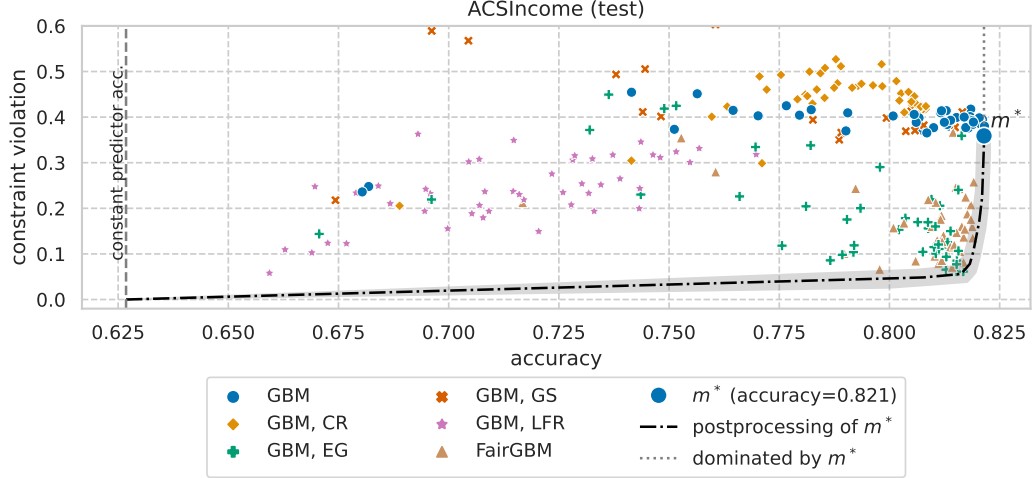

Figure A8: Fairness and accuracy test results for 300 GBM-based ML models (50 of each algorithm type) on the ACSIncome dataset. The unconstrained model with highest validation accuracy, $m^*$, is shown with a larger marker, and the Pareto frontier attainable by postprocessing $m^*$ is shown as a black dash-dot line, together with its 95% confidence intervals in shade.

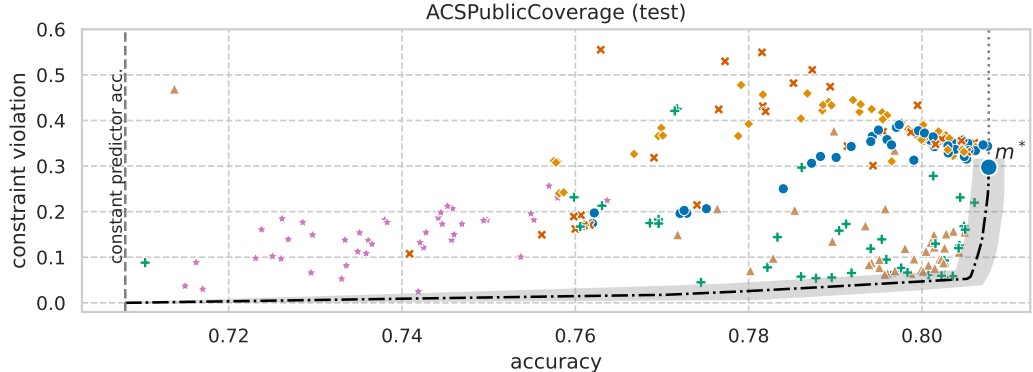

Figure A9: Fairness and accuracy test results for GBM-based ML models on the ACSPublicCoverage dataset. Model $m^*$ is of type $\langle$GBM$\rangle$ and achieves $0.808$ accuracy. See legend and caption of Figure A8 for more details.

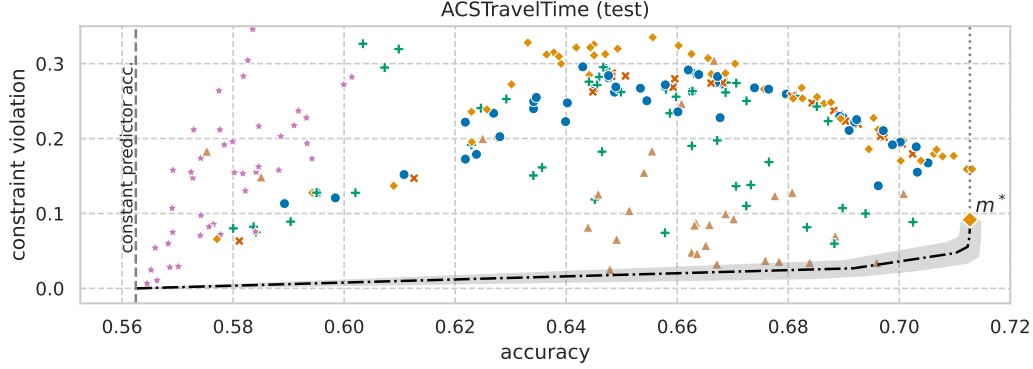

Figure A10: Fairness and accuracy test results for GBM-based ML models on the ACSTravelTime dataset. Model $m^*$ is of type $\langle$GBM,CR$\rangle$ and achieves $0.713$ accuracy. See legend and caption of Figure A8 for more details.

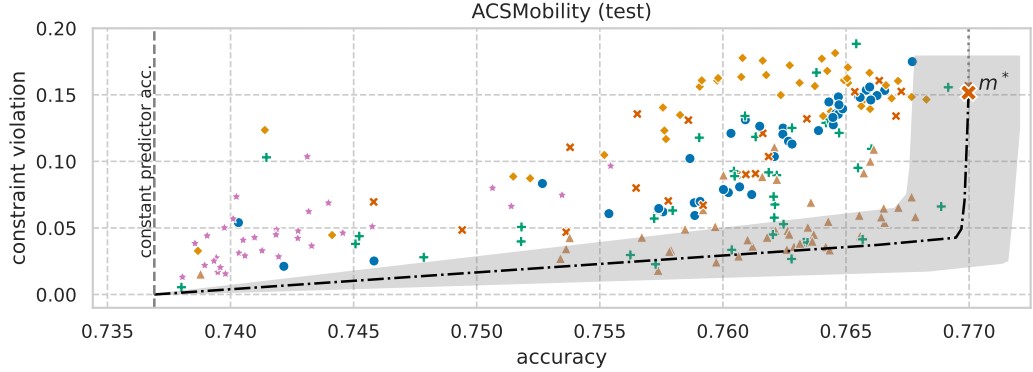

Figure A11: Fairness and accuracy test results for GBM-based ML models on the ACSMobility dataset. Model $m^*$ is of type $\langle$GBM,GS$\rangle$ and achieves $0.770$ accuracy.

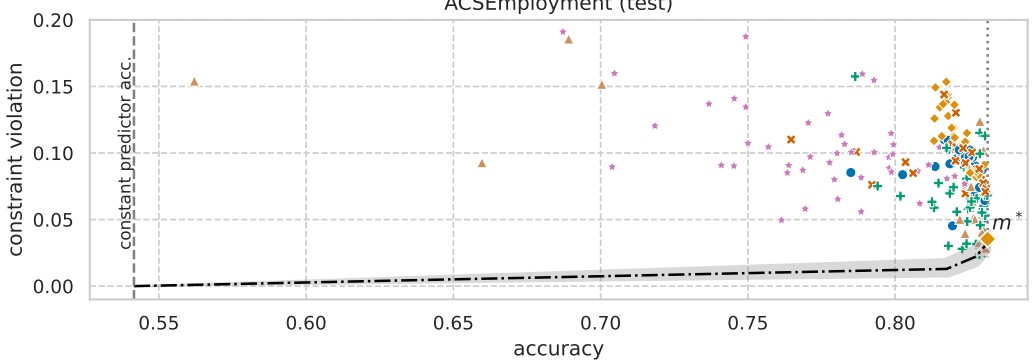

Figure A12: Fairness and accuracy test results for GBM-based ML models on the ACSEmployment dataset. Model $m^*$ is of type $\langle$GBM,GS$\rangle$ and achieves $0.831$ accuracy.

## A.3 TIME TO FIT EACH METHOD

Figure A13 shows the mean time to fit each GBM-based model on three separate datasets. The trend is clear on all studied datasets: postprocessing is a small increment to the time taken to fit the base model, preprocessing methods take longer but are still within the same order of magnitude, the FairGBM inprocessing method also incurs a relatively small increment to the base model time, while EG and GS take one to two orders of magnitude longer to fit.

For clarification, all times listed are end-to-end process times for fitting and evaluating a given model. For example, postprocessing times include the time taken to fit the base GBM model plus the time taken to solve the LP. We note that most time consumed for postprocessing simply corresponds to computing the model scores for the respective dataset where postprocessing will be fitted, while solving the LP usually takes only a few seconds. Likewise, preprocessing fairness methods include the time taken to fit the preprocessing method, the time taken to transform the input data, and the time to fit the base model. Finally, inprocessing fairness methods include only the time taken to fit the inprocessing method, as no preprocessing or postprocessing steps are required. Nonetheless, the GS and EG inprocessing methods take significantly longer than any other competing method.

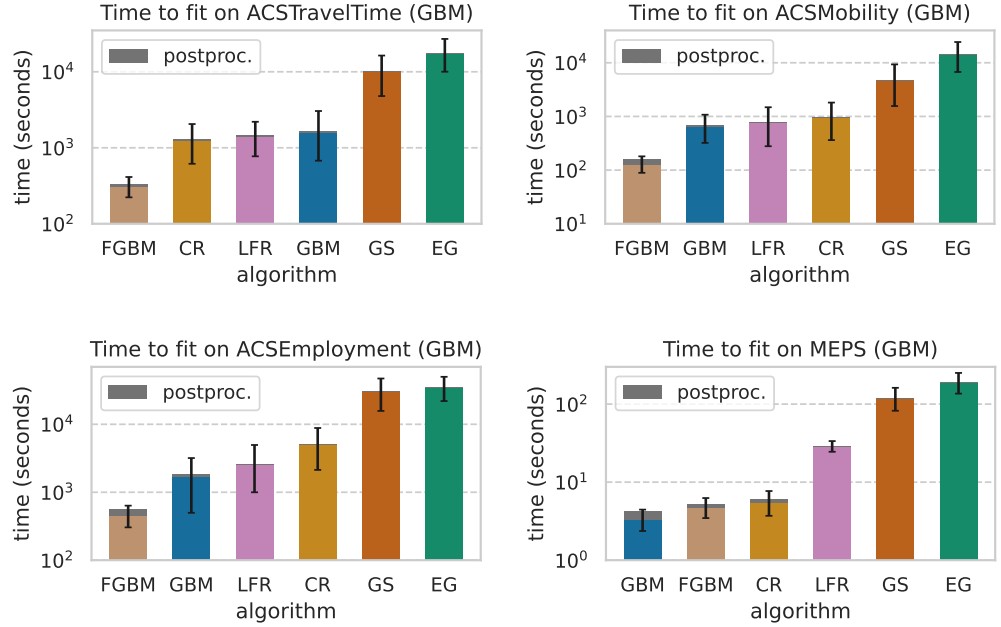

Figure A13: Mean time to fit the base GBM model and each studied fairness method on ACSTravel-Time (top left), ACSMobility (top right), ACSEmployment (bottom left), and MEPS (bottom right), with $95\%$ confidence intervals.

## A.4 EXPERIMENTS WITH BINARY SENSITIVE GROUPS

While compatibility with more than two sensitive groups is arguably essential for real-world applicability of a fairness intervention, it is common among the fair ML literature to propose and evaluate methods considering only two groups (Zemel et al., 2013; Agarwal et al., 2018; Cruz et al., 2023).

In this binary-group setting, constrained optimization methods only have to consider two constraints:

$$\left| \mathbb{P}\left[ \hat{Y} = 1 | S = 0, Y = 0 \right] - \mathbb{P}\left[ \hat{Y} = 1 | S = 1, Y = 0 \right] \right| \leq \epsilon, \qquad \triangleright \text{ FPR constraint}$$

$$\left| \mathbb{P}\left[ \hat{Y} = 1 | S = 0, Y = 1 \right] - \mathbb{P}\left[ \hat{Y} = 1 | S = 1, Y = 1 \right] \right| \leq \epsilon, \qquad \triangleright \text{ TPR constraint}$$

respectively, a constraint on group-specific FPR, and another on group-specific TPR, with some small $\epsilon$ slack. By relaxing the equalized odds problem to only two constraints we expect to provide fairness-constrained methods with the best chance at disproving the paper hypothesis.

Figure A14 (as Figure 7) shows results of applying the experimental procedure detailed in Section 2.2 to a sub-sample of the ACS datasets: only samples from the two largest sensitive groups are used (*White* and *Black*). We observe substantially lower constraint violation across the board, both for unconstrained and fairness-aware models. In fact, even unconstrained unprocessed models ($m^*$ on each plot) achieve below $0.1$ constraint violation on 4 datasets when using binary groups (all but ACSIncome), and below $0.01$ on 2 datasets (ACSMobility and ACSEmployment, see Figure A14). These results arguably discourage the use of binary sensitive groups on the ACSMobility and ACSEmployment datasets for fairness benchmarking, as very low disparities are effortlessly achieved.

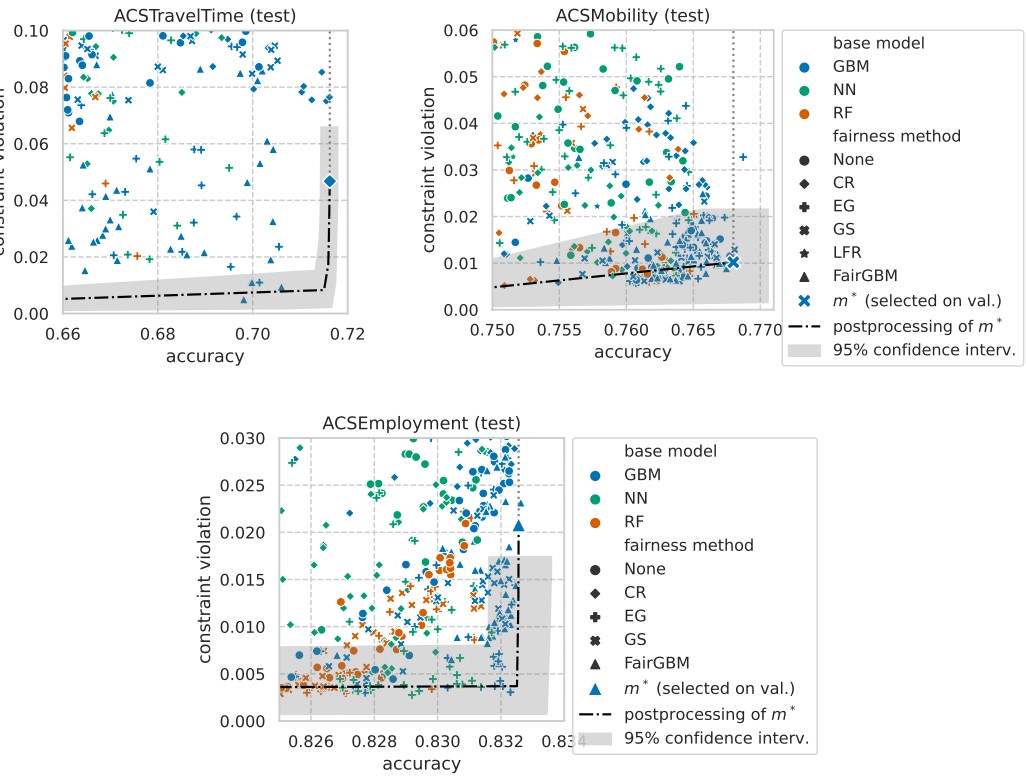

Figure A14: [**Binary protected groups**] Detailed view of the postprocessing Pareto frontier on the ACSTravelTime (left), ACSMobility (right), and ACSEmployment (bottom) datasets, when using only samples of the two largest groups (*White* and *Black*). Note the significantly reduced y axis range (constraint violation) when compared with results using four sensitive groups.

## A.5 RESULTS ON THE MEPS DATASET

The Medical Expenditure Panel Survey (MEPS) (Blewett et al., 2021) dataset consists of large-scale surveys of families and individuals across the United States, together with their medical providers and employees. MEPS collects data on the health services used, costs and frequency of services, as well as demographic information of the respondents. The goal is to predict *low* ($< 10$) or *high* ($\geq 10$) medical services utilization. Utilization is defined as the yearly sum total of office-based visits, hospital outpatient visits, hospital emergency room visits, hospital inpatient stays, or home health care visits. Exact data pre-processing is made available in the supplementary materials.[2] We use survey panels 19 and 20 for training and validation (data is shuffled and split 70%/30%) — collected in 2015 and beginning of 2016 — and survey panel 21 for testing — collected in 2016. In total, the MEPS dataset consists of 49075 samples, 23380 of which are used for training, 10020 for validation, and 15675 for testing, making it over one order of magnitude smaller than the smallest ACS dataset in our study. We use race as the sensitive attribute, with 3 non-overlapping groups as determined by the panel data: *Hispanic*, *Non-Hispanic White*, and *Non-White*.

Figure A15 shows results of conducting the experiment detailed in Section 2.2 on the MEPS dataset. We note that the variance of results is the largest among all studied datasets, as evidenced by the wide confidence intervals. This is most likely due to the small dataset size. It is also possible that the $m^*$ model on smaller datasets (such as MEPS) could produce scores that are farther from Bayes optimality than those of $m^*$ on larger datasets (such as ACS). We hope that our study motivates additional empirical work on when exactly the optimality of postprocessing breaks in practice. We recall that, although no counter-example was observed among 11 000 trained models, there are known edge-cases where postprocessing is sub-optimal (Woodworth et al., 2017). Overall, empirical results on the MEPS dataset are in accordance with those observed on the ACS datasets: the most accurate unconstrained model can be postprocessed to match or dominate any other fairness-aware model.

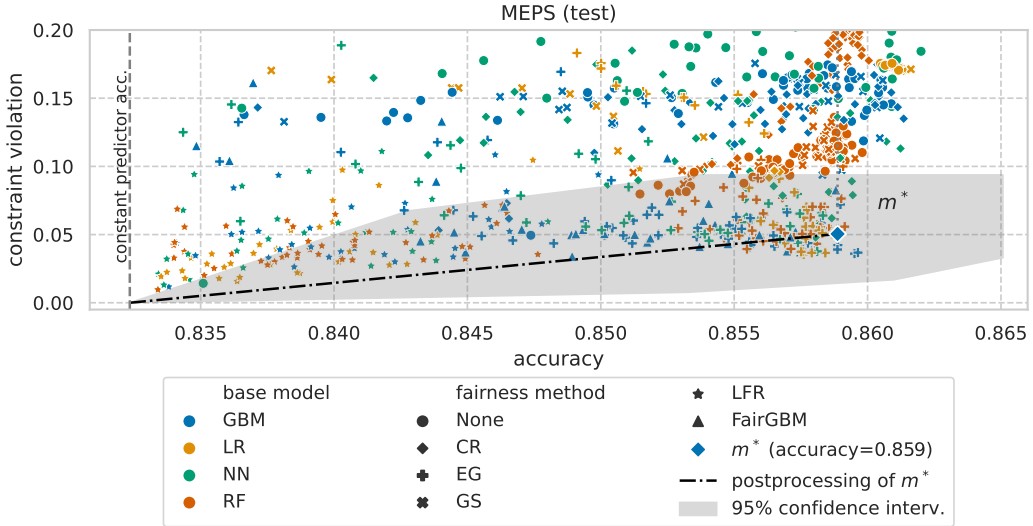

Figure A15: Detailed view of the postprocessing Pareto frontier of $m^*$ on the MEPS dataset. Note the substantial variance in results, as shown by the wide postprocessing confidence intervals.

A.6   RANKING PRESERVATION BETWEEN UNPROCESSED AND POSTPROCESSED VERSIONS

Figure A16 — akin to Figure 3 — shows real-data examples of the unprocessing-postprocessing experimental setup described in Section 2. The three plot panels show: (left) original results, (middle) results after unprocessing all models, and (right) original results with postprocessing curves overlaid.

We recall that the main experimental results (in Section 3) show that postprocessing the model with highest accuracy Pareto-dominates all other models (both fairness-aware and standard models). In this section, we present another perspective on the same empirical insight: given two specific incomparable models ($A$ and $B$), the postprocessing curve of the model with highest unprocessed accuracy will Pareto-dominate the postprocessing curve of the model with lower unprocessed accuracy. That is, while Figure 6 compares postprocessing to all other fairness interventions, Figure A16 compares postprocessing to postprocessing. In this scenario, the same empirical insight is confirmed: taking the model with highest accuracy is superior at all levels of fairness constraint violation.

In summary, when near Bayes optimality,[3] model rankings are maintained across all postprocessing relaxations, i.e., if $A^* \succeq B^*$, then $\pi_r(A) \succeq \pi_r(B), \forall r \in [0, 1]$. We know this to be true on both extremes ($r = 0 \vee r = 1$) for a Bayes optimal model (Hardt et al., 2016): it achieves optimal accuracy, and its postprocessing achieves optimal fairness-constrained accuracy. At the same time, we know this to be false on some carefully constructed counter-examples (Woodworth et al., 2017). The focus of the present work is to study whether this ranking is generally maintained in practice, on real-world data. This hypothesis is confirmed on all experiments conducted throughout the paper.

---

[3]We only compare models that are Pareto-dominant among their algorithm cohort.

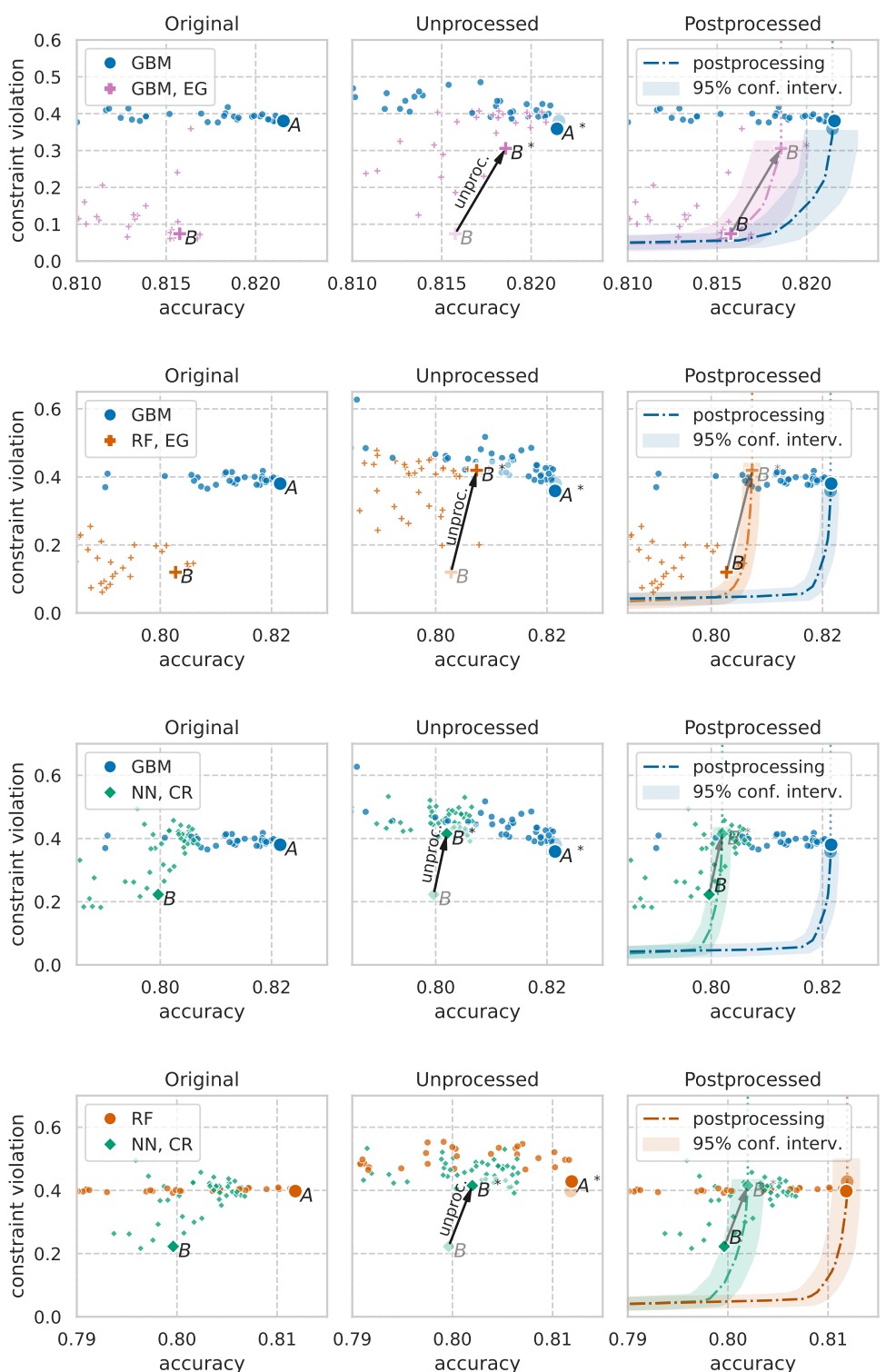

Figure A16: Comparison between postprocessing results between a variety of model pairs. Each model is selected as maximizing accuracy (model $A$) or maximizing a weighted average between accuracy and fairness (model $B$) among all models of the same algorithm cohort. Selection is performed on validation data, and results are shown on withheld test data; hence why some models may not be exactly at the Pareto frontier of their cohort. Results shown for the ACSIncome dataset.

One final noteworthy point is that unconstrained models are not significantly affected by unprocessing, occupying approximately the same fairness-accuracy region before and after optimization over group-specific thresholds (e.g., compare $A$ with $A^*$ in Figure A16). This is expected, as unconstrained learning optimizes for calibration by group (Liu et al., 2019), $P[Y = 1 | R = r, S = s] = r, \forall s \in \mathcal{S}$, which leads to the same loss-minimizing threshold for all groups (further details in Appendix C).

## A.7 UNPROCESSING VS UNCONSTRAINED LEARNING

As per Section 3, the best performing inprocessing fairness interventions are EG and FairGBM (i.e., highest Pareto-dominated area). In this section, we assess how unconstrained learning compares to unprocessing a model that was trained using either of these fairness interventions. Ideally, if enforcing the fairness constraint in-training did not hinder the learning process, we'd expect unprocessed models to approximately occupy the same fairness-accuracy region as unconstrained models.

Figure A17 shows results before and after unprocessing fairness-constrained models on the AC-SIncome dataset. Unprocessing is done on validation data, and results are shown on withheld test data. The plots show that, after unprocessing, fairness-constrained models are naturally brought to similar levels of constraint violation as unconstrained models. While overlap between unconstrained and fairness-constrained models was previously minimal or non-existent (left plots), these models form clearly overlapping clusters after unprocessing (right plots). Figure A18 shows similar results before and after unprocessing fairness-constrained models, as well as results after postprocessing unconstrained models. As evident in the plots, unprocessing brings fairness-constrained models to the high-accuracy and high-disparity region that was previously occupied solely by unconstrained models; while postprocessing brings unconstrained models to the low-disparity region previously occupied solely by fairness-constrained models. This motivates the naming of *unprocessing*, as it can be seen as the inverse mapping of postprocessing. With these plots we aim to bring attention to the interchangeability of the underlying scores produced by both unconstrained and constrained models. Whether we want to deploy a fairness-constrained or an unconstrained classifier can be chosen after model training, by postprocessing a high-performing model to the appropriate value of fairness-constraint fulfillment. Finally, postprocessing has the added advantage of better-tuned fairness-constraint fulfillment, as models that were trained in a fairness-constrained manner suffer from a wide variability of constraint fulfillment (orange markers of left-most plots).

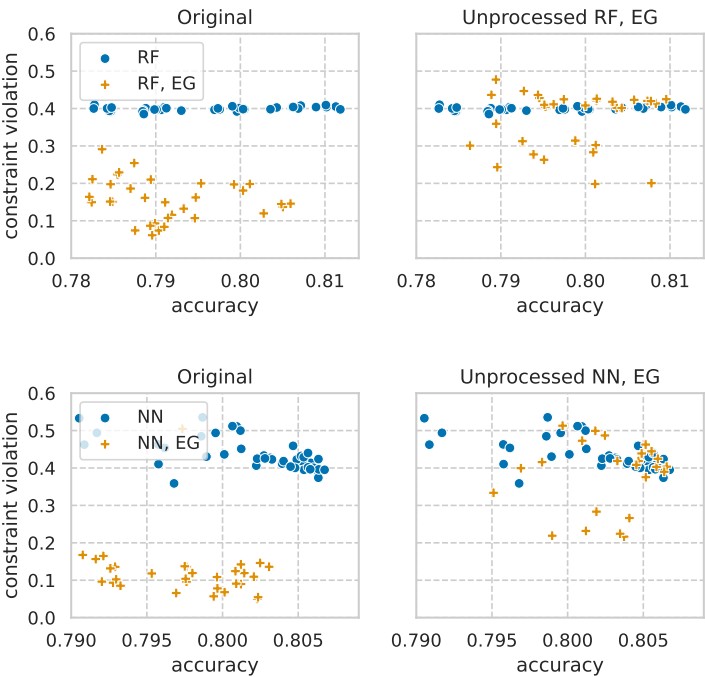

Figure A17: ACSIncome test results before (left) and after (right) unprocessing constrained models.

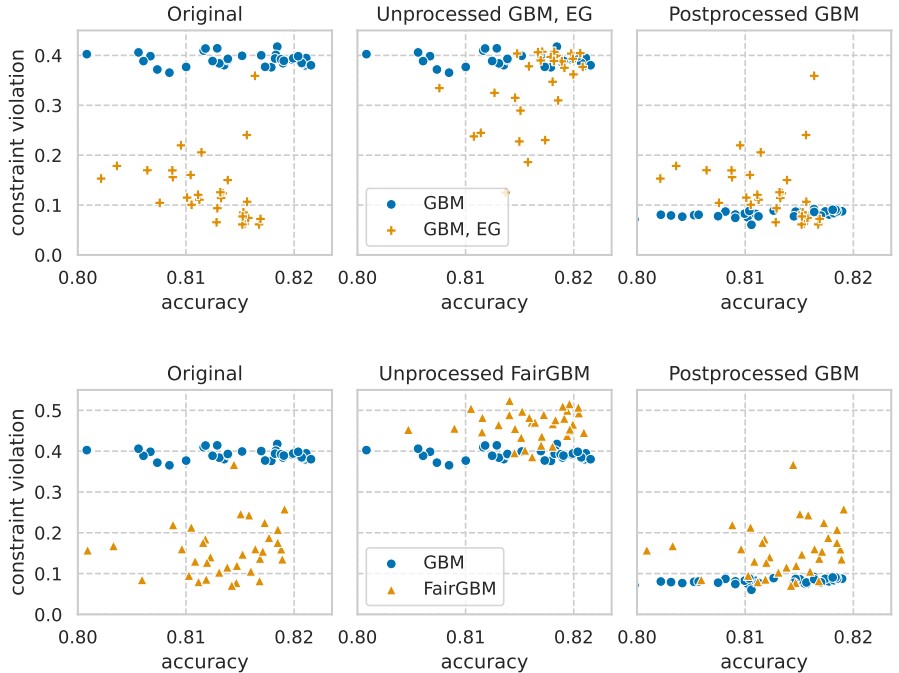

Figure A18: ACSIncome test results using GBM as the base model. *Left:* original results. *Middle:* after unprocessing fairness-constrained models. *Right:* after postprocessing unconstrained models.

## B    EXPERIMENT RUN DETAILS

All experiments were ran as jobs submitted to a centralized cluster, running the open-source `HTCondor` scheduler. Each job was given the same computing resources: 1 CPU. Compute nodes use `AMD EPYC 7662` 64-core CPUs. No GPUs were used. Memory was allocated as required for each algorithm: all jobs were allocated at least 16GB of RAM; GS and EG jobs were allocated 64GB of RAM as these ensembling algorithms have increased memory requirements.

An experiment job accounts for training and evaluating a single model on a given dataset. That is, 1 000 models were trained on each dataset (50 per algorithm type), totaling 11 000 models trained: 5 000 for the main ACS experiment using 4 sensitive groups, 5 000 for the ACS experiment using 2 sensitive groups, and 1 000 for the MEPS dataset experiment. Overall, the median job finished in 10.3 minutes, while the average job lasted for 112.0 minutes (most models are fast, but some fairness-aware models such as EG take a long time to fit, as seen in Figures 5 and A13). Compute usage was: 10 528 CPU hours for the main 4-group ACS experiment, 9 967 CPU hours for the binary group ACS experiment (Appendix A.4), and 31 CPU hours for the MEPS dataset experiment (Appendix A.5). Total compute usage was 20 526 CPU hours, which amounts to 14 days on a 64-core node. Detailed per-job CPU usage is available under folder `results` of the supplementary materials.[2]

Complete code base required to replicate experiments is provided as part of the supplementary materials, together with exact evaluation results for each trained model.[2]

## C    THRESHOLDING GROUP-CALIBRATED PREDICTORS

In this section we provide a proof for the following statement: for any classifier with group-calibrated scores (Equations 8–9), the group-specific decision thresholds that minimize the classification loss among each group all take the same value, $t_a = t_b, \forall a, b \in \mathcal{S}$, which is fully determined by the loss function, $t_s = \frac{\ell(1,0)}{\ell(1,0)+\ell(0,1)}, \forall s \in \mathcal{S}$.

*Proof.* Given a joint distribution over features, labels, and sensitive attributes $(X, Y, S)$, a binary classification loss function $\ell : \{0, 1\}^2 \to \mathbb{R}^+$, predictive scores $R = f(X)$, and binary predictions $\hat{Y} = \mathbb{1}\{R \geq t\}, t \in \mathcal{T} \subseteq \mathbb{R}$. Assume the scores $R$ are *group-calibrated* (Barocas et al., 2019), i.e.:

$$\mathbb{P}[Y = 1|R = r, S = s] = r, \qquad \forall r \in [0, 1], \quad \forall s \in \mathcal{S}, \tag{8}$$

$$\mathbb{P}[Y = 0|R = r, S = s] = 1 - r, \qquad \forall r \in [0, 1], \quad \forall s \in \mathcal{S}. \tag{9}$$

We want to minimize the expected loss among samples of group $s$, $L_s(t) = \mathbb{E}\left[\ell(\hat{Y}, Y)|S = s\right]$:

$$L_s(t) = \ell(1, 0) \cdot \mathbb{P}\left[\hat{Y} = 1, Y = 0|S = s\right] + \ell(0, 1) \cdot \mathbb{P}\left[\hat{Y} = 0, Y = 1|S = s\right], \tag{10}$$

assuming w.l.o.g. no cost for correct predictions $\ell(0, 0) = \ell(1, 1) = 0$.

We have:

$$\mathbb{P}\left[\hat{Y} = 1, Y = 0|S = s\right] = \mathbb{P}\left[\hat{Y} = 1|Y = 0, S = s\right] \cdot \mathbb{P}[Y = 0|S = s] = h_s^{\text{FP}}(t) \cdot \mathbb{P}[Y = 0|S = s],$$

$$\mathbb{P}\left[\hat{Y} = 0, Y = 1|S = s\right] = \mathbb{P}\left[\hat{Y} = 0|Y = 1, S = s\right] \cdot \mathbb{P}[Y = 1|S = s] = h_s^{\text{FN}}(t) \cdot \mathbb{P}[Y = 1|S = s],$$

where $h_s^{\text{FP}}(t)$ and $h_s^{\text{FN}}(t)$ are, respectively, the False Positive Rate (FPR) and the False Negative Rate (FNR) among samples of group $s$, as functions of the chosen group-specific threshold $t$. We can trade-off FPR and FNR by varying the threshold, leading to a 2-dimensional curve known as the Receiver Operating Characteristic (ROC) curve.

Furthermore, given the conditional density function of $R$ given $S = s$, $p_{R|s}(r)$, we have:

$$
\begin{aligned}
h_s^{\text{FP}}(t) &= \mathbb{P}\left[\hat{Y} = 1|Y = 0, S = s\right] \\
&= \mathbb{P}[R \geq t|Y = 0, S = s] \\
&= \frac{\mathbb{P}[Y = 0|R \geq t, S = s] \cdot \mathbb{P}[R \geq t|S = s]}{\mathbb{P}[Y = 0|S = s]} \\
&= \int_t^1 \frac{(1 - r) \cdot p_{R|s}(r)}{\mathbb{P}[Y = 0|S = s]} \, dr, \qquad \triangleright \text{ using calibration (Eq. 9)}
\end{aligned}
$$

$$\frac{\partial h_s^{\text{FP}}}{\partial t} = \frac{(t - 1) \cdot p_{R|s}(t)}{\mathbb{P}[Y = 0|S = s]},$$

and,

$$
\begin{aligned}
h_s^{\text{FN}}(t) &= \mathbb{P}\left[\hat{Y} = 0|Y = 1, S = s\right] \\
&= \mathbb{P}[R < t|Y = 1, S = s] \\
&= \frac{\mathbb{P}[Y = 1|R < t, S = s] \cdot \mathbb{P}[R < t|S = s]}{\mathbb{P}[Y = 1|S = s]} \\
&= \int_0^t \frac{r \cdot p_{R|s}(r)}{\mathbb{P}[Y = 1|S = s]} \, dr, \qquad \triangleright \text{ using calibration (Eq. 8)}
\end{aligned}
$$

$$\frac{\partial h_s^{\text{FN}}}{\partial t} = \frac{t \cdot p_{R|s}(t)}{\mathbb{P}[Y = 1|S = s]}.$$

The threshold $t_s$ that minimizes the group-specific loss $L_s(t)$ is a solution to $\frac{\partial L_s}{\partial t} = 0$, where:

$$L_s(t) = \ell(1, 0) \cdot h_s^{\text{FP}}(t) \cdot \mathbb{P}[Y = 0|S = s] + \ell(0, 1) \cdot h_s^{\text{FN}}(t) \cdot \mathbb{P}[Y = 1|S = s],$$

$$\frac{\partial L_s}{\partial t} = \ell(1, 0) \cdot (t - 1) \cdot p_{R|s}(t) + \ell(0, 1) \cdot t \cdot p_{R|s}(t).$$

Hence, for a group-calibrated predictor (fulfilling Equations 8–9), for any group $s \in \mathcal{S}$, the optimal group-specific decision threshold $t_s$ does not depend on any group quantities, and is given by:

$$t_s = \frac{\ell(1, 0)}{\ell(1, 0) + \ell(0, 1)}.$$

$\square$

