# OpenReview forum: "Unprocessing Seven Years of Algorithmic Fairness"
_ICLR.cc/2024/Conference — ICLR 2024 oral_

### Official Review · Reviewer_4YF6 · 2023-10-20

**Soundness:** 3 good
**Presentation:** 3 good
**Contribution:** 3 good
**Rating:** 8
**Confidence:** 3

**Summary:**

The problem the paper considers is building accurate models subject to a fairness constraint. There are many ways of building models but it is difficult to compare between different methods because a) the model performance depends on the underlying classifier and b) the models satisfy the fairness constraint up to different relaxations.

This paper seeks to solve both problems and run a large experiment on many different methods and models. They start with an approach they call "unprocessing" which takes the underlying classifier and removes the fairness constraint. In this way, different models can then reasonably be compared to each other. They then postprocess the classifiers to achieve the fairness constraint. There is an optimal way to achieve the postprocessing so this step also lets different models be compared to each other.

**Strengths:**

1. A simple way of comparing models with different fairness constraints. I hope this becomes widely adopted and used before people introduce their XYZ fairness algorithm.

2. A comprehensive evaluation of lots of models on four data sets. I especially liked two observations from their results:

* Models subject to a fairness constraint can actually achieve higher accuracy than models not subject to a fairness constraint when compared fairly (pun intended). The explanation they give is that fair training can take longer and use more resources because of the complexity in the algorithms.

* In their words:

"Crucially, postprocessing the single most accurate model resulted in the fair optima for all values of fairness constraint violation on all datasets, either dominating or matching other contender models (within 95% confidence intervals). That is, all optimal trade-offs between fairness and accuracy can be retrieved by applying different group-specific thresholds to the same underlying risk scores."

I think this is intuitively obvious and it's nice to see experimental confirmation.

3. A technical description of how to achieve relaxed parity.

**Weaknesses:**

1. I found the technical description of how to achieve relaxed parity jarring from the rest of the paper. I would have liked this section to be longer and for more explanations there. I did find the figures quite helpful in understanding it.

2. A big selling point of the paper is the extent of their experiments. I think the reason they were able to do this is because they had access to a ton of compute. All the data sets and models (I believe) are easily accessible. If this is the case, I'm not sure that "having lots of compute" is really something we should reward as a contribution.

3. I found their approach intuitively obvious: Of course given a classifier, you can vary how much it violates a reward constraint in an optimal way. So I think the contribution here would be because (it seems like) no one has done this before rather than because it is so interesting.

**Questions:**

Is there anything in my assessment you disagree with?

Have you considered putting your approach into a popular package so that researchers can quickly and easily compare their models?

---

> ### Author Response · Authors · 2023-11-20
>
> Thank you for your encouraging review and valuable feedback. We address each question in the following paragraphs.
>
> **Q2**
> > Have you considered putting your approach into a popular package so that researchers can quickly and easily compare their models?
>
> Thank you for bringing this up. We've open-sourced our implementation in a standalone python package (link to [anonymized repository](https://anonymous.4open.science/r/error-parity-8550/README.md) in the paper). Our implementation is easy-to-use and compatible with any score-based classifier (examples [here](https://anonymous.4open.science/r/error-parity-8550/examples/relaxed-equalized-odds.usage-example-folktables.ipynb)). Additionally, to reach a potentially wider audience, we have open Pull Requests to include our implementation on a popular algorithmic fairness python package (will be linked in the paper after deanonymization period), although this will expectedly still take some weeks of work to be made compatible with that package's API and structure.
>
> **W1**
>
> Thank you for raising this concern.
>
> Achieving error-rate parity was detailed by Hardt et al. (2016). The relaxed solution is more nuanced (since we can no longer rely on the intersection of all group-specific ROC curves), but the key idea is similar: using randomized thresholds allows us to access the convex hull of each group-specific ROC curve (while deterministic thresholds only allow us to access specific discrete ROC points). This makes it so the optimization domain is convex (the group-specific ROC convex-hulls), and hence easy to optimize over. While this is an efficient solution to the problem, a simple brute-force approach would also be possible (brute force implementation example [here](https://anonymous.4open.science/r/error-parity-8550/examples/brute-force-example_equalized-odds-thresholding.ipynb)).
> As linear optimization has been widely studied in the literature, we cite reference works such as Boyd and Vandenberghe (2004) for details on how to solve the LP.
> According to reviewer feedback, we were in fact torn between keeping this section in the paper or in the appendix, but as per your feedback we will maintain it in the main paper body.
>
> **W2**
>
> Thank you for raising this point. Appendix B details the infrastructure we used to conduct our experiments. Note that the run-times in Fig. 5 correspond to model training times using a single CPU-core per job (to allow for high job parallelization and a more efficient use of CPU nodes). We do believe that the variety of datasets and models tested significantly contributes to the generalizability of our findings. For full transparency, we will further detail the compute usage of our experiments in Appendix B.
>
> **W3**
>
> We agree that our approach is definitely intuitive. At the same time, the algorithmic fairness literature contains a wide range of fairness-aware inprocessing and preprocessing methods that claim to dominate results achieved by a simple postprocessing baseline. In our work, we show that this outperformance can be due to (1) comparing models at different levels of constraint violation, or (2) postprocessing a lower-performance model; both are arguably instances of unfair evaluation standards. Acting on each stage of the ML pipeline undoubtedly has specific advantages (e.g., preprocessing is potentially compatible with any downstream task/model, and can be useful when you have to hand over the data to a third party). With this meta study, we hope to have made the advantage of postprocessing clear: given accurate risk-score estimates, postprocessing can retrieve any optimal fairness-accuracy trade-off.

---

> > ### Comment · Reviewer_4YF6 · 2023-11-22
> >
> > Thanks for your detailed response!
> >
> > I am concerned about the overlap with Hardt et al. as per the review of RSrR. Nonetheless, from your response to **W1**, it sounds like your theoretical result is sufficiently different. As such, I will increase my rating to 8.
> >
> > If RSrR or the other reviewers are not persuaded that the theoretical result is sufficiently novel and different from Hardt et al., I will revise my recommendation back down to 6.

---

> > > ### Author Response · Authors · 2023-11-23
> > >
> > > Thank you for engaging with our rebuttal and for the raised score.

---

### Official Review · Reviewer_tygt · 2023-10-30

**Soundness:** 4 excellent
**Presentation:** 4 excellent
**Contribution:** 3 good
**Rating:** 6
**Confidence:** 5

**Summary:**

This work performs an extensive benchmark for 1000 models to compare the error rate disparity and accuracy trade-offs. To make a fair comparison, the constrained models, either trained with pre-processing techniques or in-processing learning constraints, are unprocessed to yield the corresponding optimal unconstrained model. Through these assessments, the authors convey a straightforward yet crucial finding: achieving fairness is best attained by training the most effective unconstrained model available and subsequently employing post-processing techniques to fine-tune the thresholds.

**Strengths:**

- I like the way the authors pose the narrative of this work. The structure is well-defined, presenting experimental details clearly.
-  I think the concept of "unprocessing" is a novel and effective method to discover the optimal unconstrained model corresponding to the constrained models.
- In general, the evaluation is solid and can provide enough insights to the practitioners.
- In my personal opinion, this paper satisfies my standard of acceptance but does not reach the rating of 8. So I would rather recommend a rating of 6.

**Weaknesses:**

- I would like to see a comparison between the real unconstrained model and the unprocessed version of the constrained model. This comparison is necessary and could enhance the claim that unprocessing can be applied to find the optimal unconstrained model.
- Section 4 is just a standard LP problem in solving Equal Odds with post-processing. It is not novel and there is no need to write down it in the main paper.
- The author has admitted that their evaluation is only applied to tabular data, with a focus on 5 different partitions of the FolkTables dataset. It would be interesting to see how the conclusions can still be generalized to tasks with rich representations.

**Questions:**

- How efficient is it to solve the LP problem? Can I just exhaustively search all the combinations of the thresholds and plot the Pareto frontiers of the fairness-accuracy trade-offs?

---

> ### Author Response · Authors · 2023-11-20
>
> Thank you for your encouraging review and valuable feedback. We address each question in the following paragraphs.
>
> **Q1**
>
> Thank you for bringing this up. The LP formulation is actually paramount to the usefulness of our method, as an exhaustive search over all threshold combinations scales exponentially with the number of groups, and scales quadratically with the number of thresholds that the underlying predictor accepts. To clarify, an exhaustive search would have to span all combinations of group-specific _randomized thresholds_ in order to access the interior of each group's ROC curve (by using deterministic single-value thresholds we can only access specific discrete points in the exterior of each group's ROC curve). That is, each group’s decision function is represented by 2 values, $\{(\underline{t}_s, \overline{t}_s) : (\underline{t}_s, \overline{t}_s) \in \mathcal{T}^2 \land  \underline{t}_s \leq \overline{t}_s\}, \mathcal{T} \subseteq \mathbb{R}$. Samples of group $s$ whose score is lower than $\underline{t}_s$ are classified negatively ($\hat{Y}=0$), samples whose score is higher than $\overline{t}_s$ are classified positively ($\hat{Y}=1$), and samples whose score is in the range $\left[\underline{t}_s, \overline{t}_s\right]$ are classified randomly by a coin toss [Hardt et al., 2016, Section 3.2].
>
> For example, a coarse search grid over deterministic thresholds of $\mathcal{T} = \{0, 0.1, 0.2, ..., 1.0\}$, includes randomized thresholds $\{ (0.0, 0.0), (0.0, 0.1), ..., (0.0, 1.0), (0.1, 0.1), (0.1, 0.2), ... \}$, a total of $\frac{\left|\mathcal{T}\right|(\left|\mathcal{T}\right|+1)}{2}$ combinations (scales quadratically with $\left|\mathcal{T}\right|$). Perhaps more importantly, if $\mathcal{A}$ is the set of randomized thresholds, the search space will span $\mathcal{A}^{|\mathcal{G}|}$ different threshold combinations, where $|\mathcal{G}|$ is the number of sensitive groups.
>
> We've implemented an exhaustive-search solver and **added an example notebook** to the examples folder in the anonymized repository linked in the paper ([link here](https://anonymous.4open.science/r/error-parity-8550/examples/brute-force-example_equalized-odds-thresholding.ipynb)). Running an extremely coarse grid of $|\mathcal{T}|=11$ thresholds on our main experiment ($|\mathcal{G}|=4$) leads to over $9\text{M}$ combinations. With our implementation, a small experiment using $|\mathcal{G}|=2$ and $|\mathcal{T}|=8$ takes 3 minutes to run over the $4356$ combinations with an exhaustive-search solver, while the LP solver takes 109ms to achieve a superior solution (because the search grid is finer).
>
> **W1**
>
> Thank you for this suggestion. We've **added these comparisons in a new Appendix A.6** to the latest paper revision. Namely, we compare fairness and accuracy results for postprocessing and fairness-constraining the same model class (e.g., FairGBM compared to postprocessed GBM), or unprocessing and unconstrained training of the same model class (e.g., unprocessed FairGBM compared to GBM).
>
> Please let us know of any further suggestions you'd find beneficial for the presentation of our work.
>
> **W2**
>
> We agree that our most significant contributions are empirical, and describing the LP is not particularly novel. However, as all our findings rely on the proposed relaxed postprocessing method, we believe that a clear definition of this optimization problem should be in the body of the paper; additionally clarifying how exactly relaxed postprocessing differs from strict postprocessing (whose LP solution is detailed by Hardt et al. (2016)). We admit that a balance between lack of context and unnecessarily detailed explanations is hard to strike, as other reviewers even suggest that this section should be lengthened.
>
> ---
>
> Thank you for your insightful recommendations. We believe the latest paper additions have definitely improved our work.

---

> > ### Comment · Reviewer_tygt · 2023-11-22
> >
> > I appreciate the authors for the detailed response and the extending experiments. I will retain my score and recommend for acceptance as indicated.

---

### Official Review · Reviewer_QHJ8 · 2023-11-01

**Soundness:** 4 excellent
**Presentation:** 4 excellent
**Contribution:** 4 excellent
**Rating:** 8
**Confidence:** 3

**Summary:**

There have been many proposals in the recent literature to train fair ML models. This paper evaluates thousands of such models, and finds that a simple postprocessing technique achieves the fairness-accuracy Pareto frontier.

**Strengths:**

This type of comprehensive benchmarking of thousands of models adds a ton of value to the algorithmic fairness literature. I think the result that a simple postprocessing step achieves the Pareo frontier is very significant. I applaud the authors for taking on this task.

**Weaknesses:**

None

**Questions:**

None

---

> ### Author Response · Authors · 2023-11-20
>
> Thank you for your encouraging review. We are very glad to know that you appreciate the significance of our work.

---

### Official Review · Reviewer_RSrR · 2023-11-01

**Soundness:** 2 fair
**Presentation:** 2 fair
**Contribution:** 2 fair
**Rating:** 6
**Confidence:** 3

**Summary:**

The paper considers the relation fairness-accuracy tradeoff. In particular, the paper considers the relation between fairness (in terms of Equalized Odds) violation and accuracy of the predictor, before and after "unprocessing", and claims based on empirical observations that any Pareto-optimal tradeoff between accuracy and empirical EOdds violation can be achieved by postprocessing.

---

**Post-rebuttal**

The authors claim that Theorem 5.6 of Hardt et al. (2016) strengthens the result of empirical studies considered in the work. It would be helpful if such discussion can be incorporated in the manuscript to help readers understand this connection. After engaging with authors and going through comments by other reviewers, I have increased my evaluation from 5 to 6.

**Strengths:**

The strength of the paper comes from the extensive empirical experiments and the efforts to present the observation (that Pareto-optimal tradeoff can potentially be achieved by postprocessing. The experiments are conducted on a relatively new data set (compared to standard baseline data sets in the literature), and the setup includes exact and relaxed EOdds (Hardt et al., 2016).

**Weaknesses:**

The weakness of the paper comes from the lack of a certain level of theoretical derivation to justify the empirical findings. The proposed term "unprocessing", as noted by authors, "roughly corresponds to the inverse of postprocessing", is more of less confusing (for reasons detailed in Section __Questions__). While one can observe from extensive empirical evaluations that Pareto-optimal tradeoffs can be achieved (setting aside numerical indeterminacy), there is a worry that the results can only provide limited insight regarding the not-clearly-motivated unprocessing procedure.

**Questions:**

__Question 1__: what is the exact relation between unprocessing and postprocessing?

Based on Hardt et al. (2016), the postprocessing strategy for EOdds is trading off True Positive Rates (TPRs) and False Positive Rates (FPRs) across different demographic groups. Such procedure is _oblivious_, in the sense that only the joint distribution $(A, Y, \hat{Y})$ are utilized in the postprocessing procedure. If this specific way of postprocessing is of interest in the paper, I am not sure how to understand the relation between unprocessing and postprocessing. I can see why authors draw an analogy between unprocessing and the inverse of postprocessing. According to Equation 1, unprocessing starts from the postprocessed $\hat{Y}$ and aims to find the unconstrained optimized predictor. How can we do that with obliviously postprocessed $\hat{Y}$? How to make sure the unprocessed predictor has a sensible mapping from input features to target variable?



__Question 2__: regarding the claim that _any_ Pareto-optimal tradeoff can be achieved by postprocessing

Follow up to Question 1, if the postprocessing is defined as in Hardt et al. (2016), it would be very helpful if authors can provide a clear characterization of the relation between unprocessing and such definition of postprocessing, so that readers can understand why unprocessing is a helpful analyzing tool to understand the importance of postprocessing. Empirical evaluations can be strengthened by some certain level of theoretical analysis to make the results and message more convincing.

---

> ### Author Response · Authors · 2023-11-20
>
> Thank you for your review and valuable feedback. We address each question in the following paragraphs.
>
> ---
> > According to Equation 1, unprocessing starts from the postprocessed Ŷ and aims to find the unconstrained optimized predictor. How can we do that with obliviously postprocessed Ŷ?
>
> Thank you for raising this question. The postprocessing procedure we propose is analogous to that  of Hardt et al. (2016), only with a partially relaxed fairness constraint. Specifically, both procedures operate on _the scores_ of a score-based predictor, by finding the optimal group-specific decision-boundary (minimal loss while fulfilling the constraint). As such, these procedures are "oblivious", in the sense that they are functions of the joint distribution of $(Y, A, R)$, and do not consider the features $X$ directly.
>
> > How to make sure the unprocessed predictor has a sensible mapping from input features to target variable?
>
> Unprocessing (or, more generally, postprocessing) does not consider the features, only the scores of the underlying predictor. If the underlying predictor does not sensibly map features to the target variable, then postprocessing that predictor will expectedly have very poor results. For this reason, to unconfound our results with the performance of the base model, we pick the predictor whose scores can achieve the highest accuracy ($m^*$).
>
> > (...) I am not sure how to understand the relation between unprocessing and postprocessing.
> >
> > (...) provide a clear characterization of the relation between unprocessing and such definition of postprocessing, so that readers can understand why unprocessing is a helpful analyzing tool to understand the importance of postprocessing.
>
> The proposed postprocessing method is solely a more general version of the method of Hardt et al. (2016) that is now compatible with relaxed fairness constraint fulfillment. We call _unprocessing_ to the specific case of postprocessing where the fairness constraint was infinitely relaxed, $r=+\infty$. That is, unprocessing boils down to minimizing each group's loss over each independent group threshold (finding which point in the group's ROC achieves minimal loss).
>
> Strict postprocessing will map an *unconstrained* score-based predictor to a *fairness-constrained* classifier. We use the term "unprocessing" to intuitevely capture the reverse procedure: mapping a *constrained* score-based predictor to a *fairness-unconstrained* classifier. Of course, either procedure can be applied to any predictor, constrained or unconstrained alike, but applying unprocessing to an unconstrained predictor will expectedly not significantly change its accuracy/fairness (discussed in the last paragraph of Sec. 2.2).
>
> To clarify with an example: given a standard unconstrained classifier (e.g., GBM), strict postprocessing will map it to the fairness-accuracy space of constrained classifiers (e.g., FairGBM); on the other hand, given a constrained classifier such as FairGBM, _unprocessing_ will do the inverse mapping, outputting a classifier that will approximately sit in the fairness-accuracy space of unconstrained GBM models.
>
> We have added a **new Appendix A.6** with detailed comparisons between unprocessed models that were trained in a fairness-constrained manner, and standard unconstrained models; as well as comparisons between postprocessed models that were trained in an unconstrained manner and inprocessing fairness-constrained models.
> Hopefully this will further clarify the motivation behind unprocessing and postprocessing.
>
> ---
>
> Thank you for your feedback. Please let us know if your questions were addressed, or of any further questions you may have.

---

> > ### Comment · Reviewer_RSrR · 2023-11-21
> > **Follow-Up on Responses**
> >
> > Thank authors for the response, and for confirming that the proposed postprocessing procedure is "analogous to that of Hardt et al. (2016), only with a partially relaxed fairness constraint." The post-/un- processing involves $(Y, A, R)$ but not $X$, where $R$ is the (potentially) continuous score instead of binary or discrete prediction itself, and $X$ are features.
> >
> > Previous results on _Near Optimality_ (Theorem 5.6, Hardt et al., 2016) already provides theoretical analysis of this setting. In particular, it is showed that if we can approximate the (unconstrained) Bayes optimal regressor well enough, then we can also construct a nearly optimal non-discriminating (in terms of Equalized Odds) classiﬁer.
> >
> > What is the contribution of the current work, as compared to the aforementioned theoretical result?
> >
> > Furthermore, since unprocessing does not consider features, can authors clarify why should we use the unprocessed predictor when we are not sure if such mapping can be attained? Specifically, how to derive prediction on new data with the unprocessed predictor?

---

> > > ### Author Response · Authors · 2023-11-22
> > > **Author's response to reviewer's response**
> > >
> > > Thank you for your response. We appreciate that your review acknowledges the strength of our "extensive empirical experiments". This is indeed our main contribution.
> > >
> > > Rest assured, we're intimately familiar with Theorem 5.6 in Hardt et al (2016). Theorem 5.6 shows that assuming a predictor is (close to) Bayes optimal, its postprocessed version is (close to) optimal among all equalized odds classifiers. This is a wonderful theoretical result, but it only strengthes the significance of our empirical results. The reason is that the theorem, of course, cannot tell you if its assumptions are met in real settings. Indeed, there is no general way of knowing what the Bayes error is on any given dataset. Likewise, there is no way of knowing if classifiers trained in real settings are close enough to Bayes optimal (in the Kolmogorov distance required by the theorem) in order for the theorem to kick in. Our empirical work shows that this is the case, thus settling a question left open for many years by the theorem you cite.
> > >
> > > You may have misunderstood the role of unprocessing in our work. Unprocessing is not a method we propose to achieve a fairness constraint. Unprocessing is a tool we use to put our empirical evaluation on a level playing field. The issue is, a priori, that one fairness method may be superior to another simply because it starts from a better unconstrained classifier. By unprocessing all methods first, we put them on a level playing field. It's a tool, both necessary and useful, for the fair evaluation of different methods. The fact that unprocessing is, intuitively speaking, the inverse of postprocessing is the whole point. This, again, does not diminish the significance of our empirical work. Rather it's a contribution we make to point out that unprocessing can be used to compare fairness methods that start from different base models. This is a useful idea that we're sure will find use in future work.
> > >
> > > We sincerely hope that you will reconsider your negative rating of our work. We think that it is entirely reasonable for an ICLR paper to be applied and empirical, rather than theoretical. Thank you very much for your efforts in carefully evaluating our work.
> > >
> > > PS: The datasets we use are widely accepted to be superior to many earlier datasets (such as German credit - 1000 data points from 1994, or UCI Adult) that were used in the area before. This case was argued in Ding et al. (2021) and researchers have since generally switched to the datasets we use. Especially given the massive scale of our empirical evaluation, it is important to have a sufficiently large dataset to make all comparisons with high confidence.

---

### Comment · Area_Chair_jrhv · 2023-11-20
**Authors, please respond to the reviews**

Dear authors: Reviewers are positive about many aspects of this submission, but they do have some concerns. Please submit your responses soon. Thank you!

---

> ### Author Response · Authors · 2023-11-20
>
> Dear AC.
> We've just submitted detailed responses to all reviewers, together with an updated paper PDF.
> The updated appendices required running some extra results that took longer than expected to finish.
> We apologize for any inconvenience.

---

### Author Response · Authors · 2023-11-20
**Revision and rebuttal**

We thank all reviewers for their valuable feedback and suggestions. We'll address each reviewer's questions below individually.

Taking the reviews into consideration, we have added a new Appendix (A.6) with detailed comparisons between unprocessed fairness-constrained models and standard unconstrained models (as well as comparisons between postprocessed unconstrained models and fairness-constrained models). We've also added further details on compute usage (Appendix B), as well as an example brute-force implementation of relaxed postprocessing for comparison (anonymized notebook [link](https://anonymous.4open.science/r/error-parity-8550/examples/brute-force-example_equalized-odds-thresholding.ipynb)). Finally, we've added a reproducibility statement as per ICLR recommendations.

---

### Meta-Review · Area_Chair_jrhv · 2023-12-15

**Metareview:**

The message of this paper is "If, however, the goal is to equalize error rates exactly or approximately, the simplest way of doing so
is optimal: Take the best available unconstrained model and optimize over group-specific thresholds."

This is an important conclusion that deserves to be widely known and debated. All reviewers are positive about the paper.

Here are three suggestions for the authors. One, the neologism "unprocessing" is cute, but it caused some misunderstanding among reviewers (fortunately, resolved). A better title would be more direct.

Two, please discuss the implications of the conclusion "If the goal is to equalize error rates [then] optimize over group-specific thresholds." When is or is not the word "fairness" appropriate for different thresholds for different groups? One real-world reason to use a complicated so-called fairness method may be that it de-emphasizes the reality that different groups have different thresholds.

Three, this paper shows that many complicated published methods in ML are in fact inferior to a previous simple method. I suggest citing papers that reach a similar conclusion about other goals in ML and related fields. Some references:

Improvements that don't add up: ad-hoc retrieval results since 1998. TG Armstrong, A Moffat, W Webber, J Zobel, 2009.

Examining Additivity and Weak Baselines. Sadegh Kharazmi, Falk Scholer, David Vallet, Mark Sanderson, 2016.

Troubling trends in machine learning scholarship. ZC Lipton, J Steinhardt, 2018.

Winner’s curse? On pace, progress, and empirical rigor. D. Sculley, J. Snoek, A. Rahimi, and A. Wiltschko, 2018.

Are we really making much progress? A worrying analysis of recent neural recommendation approaches. Maurizio Ferrari Dacrema, Paolo Cremonesi, Dietmar Jannach, 2019.

A Metric Learning Reality Check. Kevin Musgrave, Serge Belongie & Ser-Nam Lim, 2020.

**Justification For Why Not Higher Score:**

Accept (oral) is th highest.

**Justification For Why Not Lower Score:**

Based on reviewer scores, Accept (spotlight) may be better justified. However, this paper will be of interest to a large audience. Also, it is important to communicate that the research community values simplicity over complexity that is meretricious.

---

### Decision · Program_Chairs · 2024-01-16

Accept (oral)